# 'Friend or foe' and decision making initiative in complex conflict environments

**Mathew Zuparic** [1] *, **Sergiy Shelyag** [2], **Maia Angelova** [2], **Ye Zhu** [2], **Alexander Kalloniatis** [1]

**1** Defence Science and Technology Group, Canberra, ACT, Australia, **2** School of IT, Deakin University, Melbourne, VIC, Australia

\* mathew.zuparic@defence.gov.au

## Abstract

We present a novel mathematical model of two adversarial forces in the vicinity of a non-combatant population in order to explore the impact of each force pursuing specific decision-making strategies. Each force has the opportunity to draw support by enabling the decision-making initiative of the population, in tension with maintaining tactical and organisational effectiveness over their adversary. Each dynamic model component of force, population and decision-making, is defined by the archetypal Lanchester, Lotka-Volterra and Kuramoto-Sakaguchi models, with feedback between each component adding heterogeneity. Developing a scheme where cultural factors determine decision-making strategies for each force, this work highlights the parametric and topological factors that influence favourable results in a non-linear system where physical outcomes are highly dependent on the non-physical and cognitive nature of each force's intended strategy.

**Data Availability Statement:** All relevant data are within the manuscript and its Supporting information files.

**Funding:** "This research was a collaboration between the Commonwealth of Australia

## Introduction

Mathematical modelling of conflict and its wider impacts is a decades-old challenge [1]. Such models have demonstrated quantitatively descriptive capabilities—such as successfully describing the Anglo-German arms race of 1909–1913 [2]. While predictive power remains aspirational, mathematical models nonetheless enable insight into the deeper mechanisms behind real-world conflicts [3–5]. Notable examples include analytically tractable models which dictate the formation of terrorist cells [6], analysing turbulence phenomena to understand the historical statistics of warfare [7], and applying atomic and chemical interactions to model cooperation and competition behaviour in social animals [8]. Taking inspiration from previous mathematical studies, the present work demonstrates the potential of a mathematical model to understand the thresholds separating victory and defeat between two adversaries engaging with a local population where strategic intent may not eventuate into the planned outcome due to a number of competing influencing factors. We reveal the strategies which lead to unambiguous victories of either force due to population support. In the event that both forces pursue the same strategy, we analytically highlight additional decision-making and topological properties that lead to favourable outcomes when altered.

(represented by the Defence Science and Technology Group) and Deakin University through a Defence Science Partnerships agreement. This research has been undertaken with the assistance of resources and services from the National Computational Infrastructure (NCI), which is supported by the Australian Government, and from the OzSTAR national facility at Swinburne University of Technology. The funders had no role in study design, data collection and analysis, decision to publish, or preparation of the manuscript".

**Competing interests:** The authors have declared that no competing interests exist.

## Problem statement

We introduce a model which seeks to explain the impact of adversarial forces pursuing decision-making strategies in relation to the choices of a non-combatant population caught up in the conflict. Thus three parties are involved here. Tragically, conflict rarely ever stays within the confines of the battlefield, often spilling into the wider population [9]. In response to this, there are many historical instances where opposing forces acknowledge the wider population as an actor with agency in the conflict. Examples include the British army cooperating with Spanish guerilla fighters against the French in the Napoleonic Wars [10]. More recently, the experience of 'western' forces in Iraq and Afghanistan where local governments rarely implemented the reforms requested/demanded by either external force or insurgent groups, unless those changes served their interests, and those of the wider population, often resulting in a lack of trust and support due to forces not appreciating nuanced decision-making priorities of the population [11].

## Mathematical motivation

We mathematically codify this phenomenon, where support from a population is gained by either force respecting the decision-making initiative of the population—with follow-on benefits in combat effectiveness. While less emphasised than other aspects, this is inferred in counter-insurgency doctrines of both the US Army [12] and NATO [13]. Though subsequent historical analysis may eventually answer differently, some analyses suggest that recent strategic changes in this context may be a consequence of inconsistency in the application of such approaches within the challenge of balancing multiple goals rather than a flaw in such doctrine itself [14].

One of the counterbalancing goals is the traditional one, that a combat force must remain ahead of their adversary's decision-making to maintain combat effectiveness in the classical manoeuvrist sense [15]. This naturally leads to a tension for each force's strategy: each seeks to strike the best balance between maintaining population support, and combat initiative. To explore this intrinsically non-linear phenomenon, the mathematical model introduced here combines a number of archetypal models from the wider literature inspired by multi-network [16] and complex systems modelling [17] to represent each different aspect. The interactions between these elements are illustrated in Fig 1.

The intrinsic decision-making of agents in Fig 1 is represented by the networked Kuramoto-Sakaguchi model [18] of phased oscillators, $\theta_i \in \mathbb{S}^1$:

$$\dot{\theta}_i = \omega_i - \sum_{j \in \mathcal{A}} \mathcal{A}_{ij} \sin(\theta_i - \theta_j - \phi_{ij}), \ \ i \in \mathcal{A}, \tag{1}$$

for natural frequency $\omega_i \in \mathbb{R}$, adjacency matrix $A$, and frustration $\phi \in \mathbb{S}^1$. Since its inception [19], Eq (1) has been applied to explore the onset of synchronisation in various physical and biological systems [20, 21]. More recent studies have applied generalised Kuramoto models to explore social, and socio-technical systems, which combine human decision-makers with technological artefacts/agents. Examples include the modelling of staff decision cycles within military headquarters [22], the conformists-and-contrarians [23] and opinion dynamics [24] models which explore differing ideologies and opinions of networked agents. The Kuramoto model's potential to represent decision-making processes follows the mapping between the cyclic nature of phased-oscillators dynamics, and that of human decision-making models in cognitive psychology [25], Boyd's observe-orient-decide-act (OODA) loop of business/military strategy [15] and human factors literature [26].

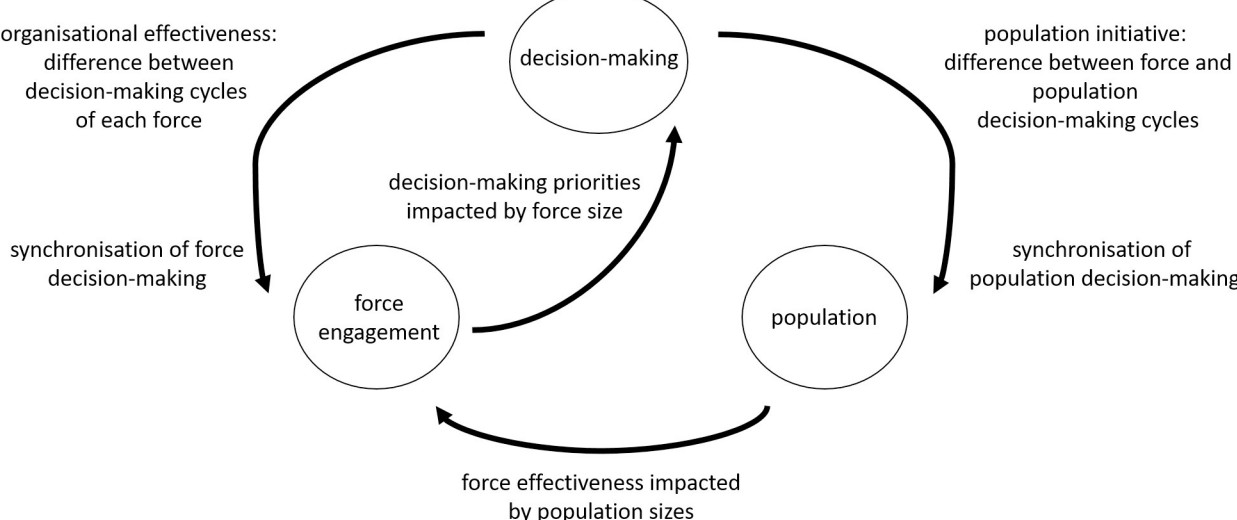

**Fig 1. Macroscopic view of each model component—Decision-making, population and force engagement—And the feedback mechanisms between them.**

Force engagement in Fig 1 is represented via the *aimed fire* Lanchester model [27]

$$\dot{B} = -\kappa_{RB} \cdot R, \quad \dot{R} = -\kappa_{BR} \cdot B, \tag{2}$$

where $B$ and $R$ in Eq (2) correspond to the (numerical) strength of the opposing Blue and Red forces, and parameters $\kappa_{BR}, \ \kappa_{RB} \in \mathbb{R}_+$ represent Blue and Red's rates of effectiveness against their adversary [28]. An equally valid variant of the Lanchester equations is the *area/unaimed fire* model [29]:

$$\dot{B} = -\kappa_{RB} \cdot B \cdot R, \quad \dot{R} = -\kappa_{BR} \cdot B \cdot R, \tag{3}$$

where combat strength depends on force size, multiplied by the density of the targets at the opposing force. As the fire is not aimed in this variant, the likelihood of acquiring an adversarial target depends on the target size. This current work applies the aimed fire Lanchester model given in Eq (2), as both Blue and Red forces are assumed to take care in the vicinity of non-combatants so as not to lose population sentiment due to accidental civilian casualties. Asymmetric Lanchester models which combine both Eqs (2) and (3)—also known as the Deitchman model [30]—have been used to model *guerrilla warfare*, relevant in scenarios where the population directly participates in the conflict. Previous studies have attempted to validate Lanchester outputs to World War II data-sets [31], and the recent Syrian civil war [32]. Recent generalisations of the Lanchester equations for heterogeneous forces include MacKay's mixed forces model [33, 34] and a fully networked Lanchester model [35].

Finally, population in Fig 1 is represented by the Lotka-Volterra model [36]

$$\dot{g}_{prey} = g_{prey}\big(\kappa_{grow}^{prey} - \kappa_{decay}^{prey} \cdot g_{pred}\big), \quad \dot{g}_{pred} = g_{pred}\big(\kappa_{grow}^{pred} \cdot g_{prey} - \kappa_{decay}^{pred}\big), \tag{4}$$

where $g_{prey}$ and $g_{pred}$ in Eq (4) correspond to the number of prey and predator species, and parameters $\kappa_{grow}$ and $\kappa_{decay} \in \mathbb{R}_+$ represent each species rate of growth and decay, respectively [37]. The Lanchester system is an adaptation of Eq (4), thus its application to describe populations in a conflict is not unnatural.

The feedback indicated in Fig 1 presents a bounded view of the system, where feedback in a more detailed setting would act both ways, and in an all-to-all arrangement. For instance, strategic decision-making would typically respond to the first-order effects due to population sentiment. The decision to limit the model as per Fig 1 is deliberate, as it was important to have enough detail so that the model could be used to explore the non-trivial impact that decisions and strategy relating to non-combatants have in combat, all-the-while bounding the complexity so that important model behaviours could be understood analytically. Refer to [38–40] as clear examples of authors who have explored the impact of complexity trade-off in model construction.

Various components of the model in Fig 1 have been considered in conflict modelling. Adversarial decision-making was explored in [41] for two networks, and in [42] for three where the tension between each force's strategy was highlighted. Distributed decision-making coupled with engagement was explored in [43]. Another variation of three groups in conflict environments using non-trophic/symbiotic Lotka-Volterra generalisations, where the third group represented external (for example, humanitarian) agencies, was explored in [44–46]. In [47, 48] the impact of influence and support from a host population, as the third party in the conflict, on engagement outcomes was analysed, and [5] explored the impact of recruitment policies on multi-party engagements. Lanchester-type models have also been used to study competitive behaviour between eusocial insects [49, 50], explore evolution success in social species [51], predict the efficacy of cancer treatment [52, 53], and understand agent success in real-time-strategy games [54].

In this paper, first, we provide the model definition, as well as typical examples of model behaviour, demonstrating how mechanisms between model components impact outcomes. Then we provide an examination of the impact of each force's chosen decision-making strategy on model outcomes. We consider a model approximation to explore the sensitivity of adversarial outcomes to the decision-making dynamics. Finally, we re-interpret model behaviours in the context of how decision-making strategies affect outcomes and illustrate how model behaviours demonstrate conceptual similarities with observations from modern conflicts.

## Model definition and behaviour

### Decision-making dynamics

Eq 1 defines the decision-making component of the model with the three parties, where the adjacency matrices and frustrations can be expressed via:

$$
\mathcal{A} = \begin{pmatrix} \sigma_B \cdot \mathcal{B} & \zeta_{BG} \cdot \mathcal{I}^{BG} & \zeta_{BR} \cdot A(B) \cdot \mathcal{I}^{BR} \\ \zeta_{GB} \cdot \mathcal{I}^{GB} & \sigma_G \cdot \mathcal{G} & \zeta_{GR} \cdot \mathcal{I}^{GR} \\ \zeta_{RB} \cdot A(R) \cdot \mathcal{I}^{RB} & \zeta_{RG} \cdot \mathcal{I}^{RG} & \sigma_R \cdot \mathcal{R} \end{pmatrix},
\tag{5}
$$

$$
\phi = \begin{pmatrix} 0 & \phi_{BG} & \phi_{BR} \\ 0 & 0 & 0 \\ \phi_{RB} & \phi_{RG} & 0 \end{pmatrix}.
\tag{6}
$$

Here, $\theta = (\theta^B, \theta^G, \theta^R) \in \mathbb{S}^1$, are time-dependent phases for $\mathcal{B}$, $\mathcal{G}$ and $\mathcal{R}$ populations, respectively, each with intra-network coupling constants $\sigma \in \mathbb{R}_+$. Coupling constants across networks are given by $\zeta \in \mathbb{R}_+$, where we assume symmetric inter-adjacency matrices, *i.e.*

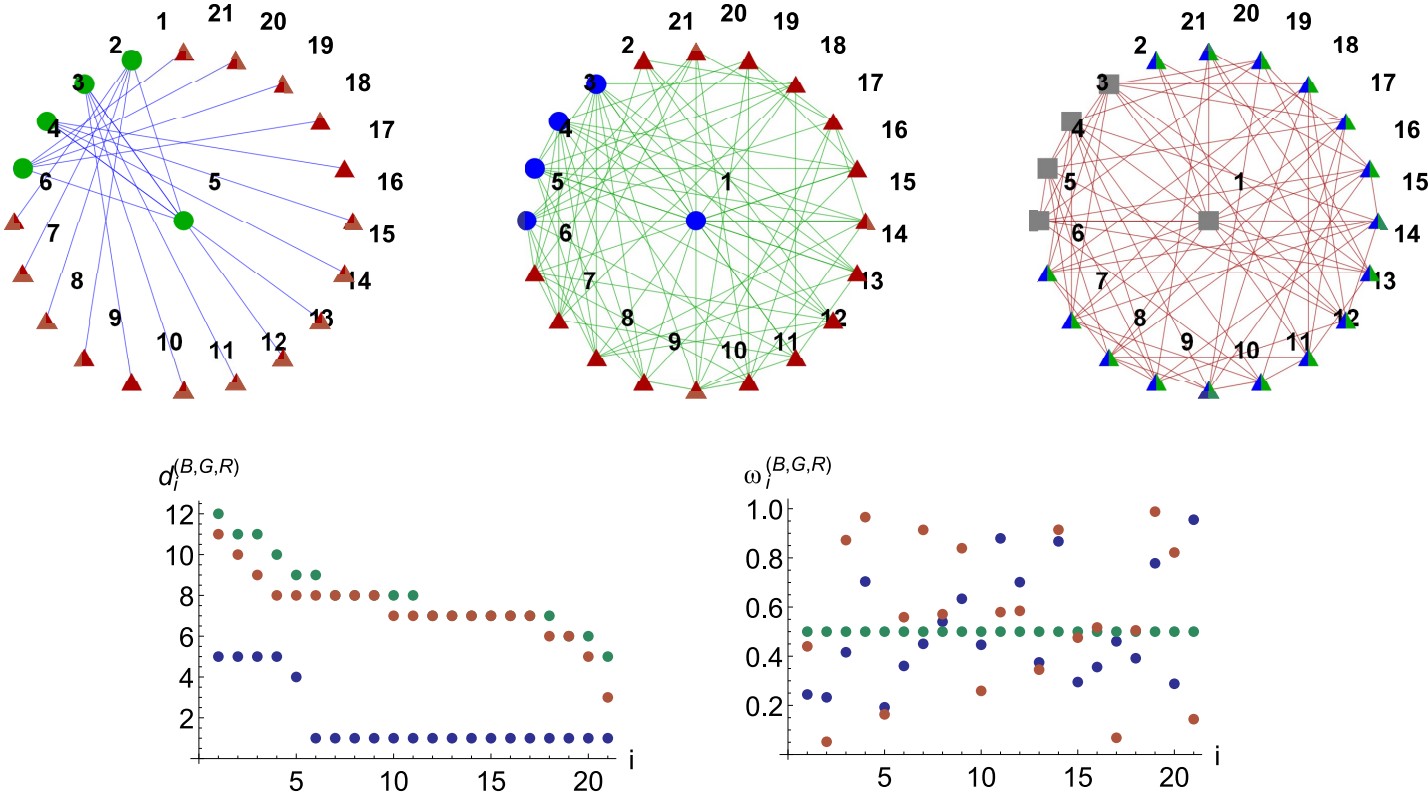

**Fig 2. Top row gives the Blue (hierarchy), Green (Watts-Strogatz) and Red (Erdös-Rényi) decision-making networks, $\mathcal{B}$, $\mathcal{G}$ and $\mathcal{R}$, used throughout this work.**
Nodes with common shapes and labels share links with other networks, indicated by the node colour. For instance, green circles labeled 1–5 in the Blue network share links with blue circles in the Green network with corresponding labels. Thus $|\mathcal{I}^{BG}| = 5$ and $|\mathcal{I}^{BR}| = |\mathcal{I}^{GR}| = 16$. The grey squares labeled 1–5 in $\mathcal{R}$ are not connected to either $\mathcal{B}$ or $\mathcal{G}$. The bottom row displays each labeled node's internal degree (left) and natural frequency (right).

$\mathcal{I}^{(XY)} = \mathcal{I}^{(YX)}$: meaning that decision-making agent interaction is symmetric, though this can be generalised. Here, $A(R)$ and $A(B)$ are attenuation terms for the Red and Blue respectively—providing feedback from conflict to decision-making. Notably, the dynamics of the conflict and population model components—defined shortly—are equivalently determined by feedback from agent decision-making. The specific network topologies and natural frequencies used for illustrative purposes throughout this work are given in Fig 2, which were the topologies considered in [42], and are provided as supplementary comma separated variable files.

Frustration parameters $\phi \in \mathbb{S}^1$ are key considerations in this work, representing an intended lag in the oscillator states between networks. This is interpreted as the *strategy*: representing one network's decision-making intent towards another. Whether the non-linear dynamics permit that strategy to be realised is the main question addressed. In the absence of making $\phi$ adaptive [55], the choice to change it sits outside the model. This will be important in how we explore the space of outcomes of the dynamics. Table 1 summarises the variables and parameters introduced in this Section.

## Force and population dynamics

Consider two opposing forces, *Blue* and *Red*, affecting each other akin to the directed-fire Lanchester attrition interaction in Eq (2). $B$ and $R$ represent their respective time-dependent force strengths. Complexity is added to this interaction by considering the impact of a non-combatant *Green* population of size $G$, composed of three sub-populations, $G = g_B + g_R + \Gamma$; $g_R$ are Red

**Table 1. Summary of the force/population and decision-making inspired variables and parameters defined in this section, and their physical interpretations.**

| expression | name | interpretation |
|---|---|---|
| $\{\theta^B, \theta^G, \theta^R\}$ | phase | decision state of agent |
| $\{\mathcal{B}, \mathcal{G}, \mathcal{R}\}$ | adjacency matrix | internal decision-making network |
| $\omega^X$ | natural frequency | decision-speed of $\mathcal{X}$ agent in isolation |
| $\sigma_X$ | intra-network coupling | intensity of intra-agent interaction in $\mathcal{X}$ |
| $\mathcal{I}^{XY}$ | inter-adjacency matrix | topology between $\mathcal{X}$ and $\mathcal{Y}$ decision-makers |
| $\zeta_{XY}$ | inter-network coupling | intensity of $\mathcal{X}$'s decision-making interaction with $\mathcal{Y}$ |
| $\phi_{XY}$ | frustration | $\mathcal{X}$'s strategy/intent towards $\mathcal{Y}$ |
| $\{B, R\}$ | force strength | size of opposing forces |
| $\{\Gamma, g_B, g_R\}$ | population | size of non-combatant sub-populations |
| $\alpha_{F_1 F_2}$ | lethality | force-$F_1$'s effectiveness against force-$F_2$ |
| $\alpha_{GF}$ | population sentiment | rate that population can change between neutral and $F$-supporters |
| $\kappa_{F_1 F_2}$ | physical effectiveness | efficacy of $F_1$'s physical capabilities applied to $F_2$ |
| $\kappa_{GF}$ | willingness/inertia | inherent enthusiasm/apathy for population to support, or rescind support from, $F$ |
| $\Omega_{F_1 F_2}$ | organisational effectiveness | efficacy of $F_1$'s decision-making capabilities applied to $F_2$ |
| $\Omega_{GF}$ | population initiative | population support for $F$ due to decision-making dynamics |

supporters, $g_B$ are Blue supporters, and sub-population $\Gamma$ maintain neutrality. The following formulation is a generalisation of the supporter model defined in [48].

The Lanchester interaction between Blue and Red is given by

$$\dot{B} = -\alpha_{RB} \cdot f(g_R, g_B) \cdot R \cdot \mathcal{H}(B), \quad \dot{R} = -\alpha_{BR} \cdot f(g_B, g_R) \cdot B \cdot \mathcal{H}(R), \tag{7}$$

$$\text{where } f(x, y) = \frac{x}{y + \epsilon}, \tag{8}$$

with the lethality coefficients $\alpha_{BR}$ and $\alpha_{RB}$ having units $[time]^{-1}$. Note that there are no growth terms; the model is therefore formulated within a context of a campaign where the logistics of force deployment into a theatre have been completed. Both expressions in Eq (7) are modulated by the term $f$, which factors in the impact of both supporter populations, $g_B$ and $g_R$. The second argument in $f$ is the size of the supporting population, and serves to decrease the adversary's effectiveness against the supported force. Correspondingly, the first argument in $f$ is the size of the detracting population who increase the adversary's effectiveness against the supported force. The addition of $\epsilon \ll 1$ in the denominator avoids any pathological behaviour, when the corresponding population in the denominator nears zero, and has the physical interpretation of a 'standing population' that the detractors wish to maintain, however small [48]. Furthermore, $\mathcal{H}$ in Eq (7) is a sufficiently smooth Heaviside-step-like function which ensures that force strengths remain positive. For this work we employ

$$\mathcal{H}(x) = 0.5\{\tanh[\epsilon^{-1}(x - 4\epsilon)] + 1\}, \text{ where } \epsilon = 10^{-6}. \tag{9}$$

The lethality coefficients $\alpha_{BR}$ and $\alpha_{RB}$ in Eq (7) are dynamic, and depend on outputs from the decision-making component.

The dynamics amongst sub-populations of Green is given by,

$$\dot{g}_R = \alpha_{GR} R \cdot g_R \cdot \Gamma \cdot \mathcal{H}(B), \;\; \dot{g}_B = \alpha_{GB} B \cdot g_B \cdot \Gamma \cdot \mathcal{H}(R), \tag{10}$$

$$\Gamma + g_R + g_B = G = const, \tag{11}$$

where $\alpha_{GR}$ and $\alpha_{GB}$ in Eq (10) are referred to as population sentiment and have units of [*population*$^2 \cdot$ *time*]$^{-1}$. Eq (10) gives the rate of change of sub-populations $g_R$ and $g_B$, both affected by the multiplication of three population sizes. Though $G$ is conserved in Eq (11), this can be generalised [48]. The population sentiment terms in Eq (10), $\alpha_{GR}$ and $\alpha_{GB}$, are derived from the decision-making model and explained shortly.

## Required outputs for feedback mechanisms

Local synchronisation denotes the extent to which all nodes of a particular sub-network are phase-locked, i.e. $\theta_i = \theta_j \, \forall \, i, j \in \mathcal{B}, \; \mathcal{G},$ or $\mathcal{R}$ exclusively. It is measured via the order parameter

$$O_X(t) \equiv \frac{\left| \sum_{j \in \mathcal{X}} e^{i\theta_j^X} \right|}{|\mathcal{X}|}, \;\; X \in \{B, G, R\}, \;\; \mathcal{X} = \{\mathcal{B}, \mathcal{G}, \mathcal{R}\}. \tag{12}$$

Here $|\mathcal{X}|$ represents the number of nodes of the graph. The range of $O$ is between zero and unity, with values close to unity representing coherence in decision-making cycles. Intuitively, the order parameter represents the organisational effectiveness of either force, and the relative cohesion of the decision-making of the non-combatants.

We also consider the global phases (or centroids) for agents in each network, denoted by $\Theta_X$, given by,

$$\Theta_X = \frac{1}{|\mathcal{X}|} \sum_{i \in \mathcal{X}} \theta_i^X. \tag{13}$$

Importantly, the difference between global phases is designated $\Delta$ with:

$$\Delta_{XY} \equiv \Theta_X - \Theta_Y = -\Delta_{YX}. \tag{14}$$

## Feedback: Decision cycles to force dynamics

Inspired by [43], lethality in combat is a product of components of physical ($\kappa$) and organisational ($\Omega$) effectiveness, through the factorisation

$$\alpha_{BR} = \kappa_{BR} \cdot \Omega_{BR}, \;\; \alpha_{RB} = \kappa_{RB} \cdot \Omega_{RB}. \tag{15}$$

The physical effectiveness is a constant positive parameter, $\kappa \in \mathbb{R}_+$, which controls the timescale of combat compared to the decision-making dynamics. Organisational effectiveness is generated dynamically via

$$\Omega_{BR} = O_B \left( \frac{1 + \sin\Delta_{BR}}{2} \right), \;\; \Omega_{RB} = O_R \left( \frac{1 + \sin\Delta_{RB}}{2} \right). \tag{16}$$

consisting of the multiplication of two dynamic quantities—$O$ and $(1 + \sin\Delta)/2$—which both vary between zero and unity. Hence, each side gains lethality by maintaining good intra-network synchronisation, and staying ahead (maximally $\pi/2$) of their adversary's collective decision phase, implementing the decision advantage approach of Boyd [15]. We have also assumed that 'ahead' means a phase oscillator is in the range $(0, \pi)$ and 'behind' is $(-\pi, 0)$. This

is somewhat arbitrary but with greater phenomenology around such cognitive processes, these functions are straightforwardly adjusted.

## Feedback: Decision cycles to population dynamics

Population sentiment coefficients, $\alpha_{GB}$ and $\alpha_{GR}$, determine whether population support for either force grows ($\alpha > 0$) or declines ($\alpha < 0$). Similar to lethality, sentiment is a product of a constant willingness/inertia term ($\kappa$), and a time-dependent initiative term ($\Omega$) via

$$\alpha_{GB} = \kappa_{GB} \cdot \Omega_{GB}, \;\; \alpha_{GR} = \kappa_{GR} \cdot \Omega_{GR}. \tag{17}$$

The constant factor $\kappa \geq 0$ represents the inherent enthusiasm of the population changing their support for either force, controlling the timescale of population dynamics compared to decision-making. Population initiative is defined by

$$\Omega_{GB} = O_G \sin\Delta_{GB}, \;\; \Omega_{GR} = O_G \sin\Delta_{GR}, \tag{18}$$

consisting of the multiplication of two dynamic quantities—$O_G$ and $\sin\Delta$. The order parameter $O_G$ means that the rate of change of support for either force amongst the non-combatants is increased by good synchronisation in the decision-making cycles of the population. Polarisation ($O_G \ll 1$) will occur when members form divided echo-chamber groups and are exposed to selective information [56].

The terms $\sin\Delta_{GB}$ and $\sin\Delta_{GR}$, mean that the direction of the change in support is dependent on the relative position of each force's decision-making cycles with respect to that of the non-combatants'. Thus, consistent with the phenomenology described in the introduction, a green sub-population, $g_F$, for $F \in \{B, R\}$, will increase support for that force if they are granted decision-making initiative, *i.e.* $\Delta_{GF} \in (0, \pi)$. This corresponds to the force $F$ fostering a positive relationship with the non-combatant population, and prioritising their best interests from the cultural perspective of the population members, not from the cultural perspective of either force [11]. Such forces would possess adequate training, encourage meaningful engagement with population leaders and facilitate their role in decision-making [57].

## Feedback: Combat to decision cycle dynamics

Combat outcomes affect the decision-making dynamics, as originally considered in [43], via the attenuation terms, $A$, in Eq (5). The choice of attenuation is context-dependent. This work addresses the case where forces that suffer losses focus less on staying ahead of their adversary's decision making via

$$A(F) = F/F_0. \tag{19}$$

Less focus on adversarial decision-making potentially enables forces greater concentration on pursuing their strategy/intent towards the population—either enabling or restricting non-combatant decision-making initiative. Table 2 summarises the feedback mechanisms.

## Constraining the space of strategy options

In general, each side must explore a four-dimensional strategy space spanned by the variables $\{\phi_{BR}, \phi_{BG}, \phi_{RB}, \phi_{RG}\}$. We consider a scenario where Blue and Red are not gifted with arbitrary computational power, so must conduct a trade-off in strategy between their adversary and the host population within a constrained subspace of possible strategies.

**Table 2. Summary of feedback mechanisms from decision-making to force and population modelling components, and their physical interpretations.**

| expression | name | interpretation |
|---|---|---|
| $O_X$ | order parameter | measure of synchronisation of decision-making agents in $\mathcal{X}$ |
| $\Theta_X$ | global phase | average phase of agents in $\mathcal{X}$ |
| $\Delta_{XY}$ | global phase difference | difference between averages of decision-making cycles of networks $\mathcal{X}$ and $\mathcal{Y}$ |
| $A(F)$ | attenuation | change in decision-making priorities due to changes in force size $F$ |
| $f(g_1, g_2)$ | modulation | impact of population supporters/detractors on force effectiveness |

To enable a low dimensional exploration and understanding of how each force's choice in strategy affects the combat outcomes, we employ a constraint defined by

$$\phi_{BR} = \mu_B \pi, \ \ \phi_{BG} = (\mu_B - \mu_B^*)\pi \tag{20}$$

$$\phi_{RB} = \mu_R \pi, \ \ \phi_{RG} = (\mu_R - \mu_R^*)\pi \tag{21}$$

in terms of renormalised frustrations $\mu_B, \mu_R$. Thus Blue explores options along a one-dimensional line within $\phi_{BR}, \phi_{BG}$ and similarly for Red; this leads to an exploration of a two-dimensional sub-space of the strategy space. The fixed positive parameters $\mu_B^*$ and $\mu_R^*$ in Eqs (20) and (21) represent a candidate trade-off threshold in the intended decision-making strategy. The choice of explored subspace may reflect cultural differences between the force $F$ and the population, namely how much of the parameter exploration allows for options ceding the initiative to the population. Thus, smaller values of $\mu_F^*$ reflect a significant cultural difference between the force $F$ and the population. Contrastingly, a value of $\mu_F^* = \frac{1}{2}$ equates to the force seeking to equally explore both advantage over the adversary and population initiative—a scenario which may be unrealistic if a significant cultural difference exists between the force and the population.

To illustrate this, we consider an intermediate scenario:

$$\mu_B^* = \mu_R^* = \frac{1}{3}. \tag{22}$$

This partitions $F_1$'s choice of $\mu_{F_1}$ into three archetypal decision-making strategies, defined in Table 3. Thus forces pursuing an intended mixed strategy have both $\phi_{F_1 F_2} > 0$ and $\phi_{F_1} G < 0$, meaning that they are trying to stay ahead of their adversary's decision-making, whilst also affording initiative to the population. Forces pursuing an intended population strategy seek to grant greater initiative to the population ($\phi_{F_1} G < -\pi/3$), at the cost of trailing their adversary's decision-making ($\phi_{F_1 F_2} < 0$). Contrary to this, the pursuit of an adversarial strategy sees the force prioritise staying ahead of their adversary's decision-making ($\phi_{F_1 F_2} > \pi/3$), to the detriment of the initiative granted to the population ($\phi_{F_1} G > 0$). Whether the nonlinear dynamics (for example, instabilities) permit such intended strategies to be viable is revealed via the mathematical model.

**Table 3. Summary of archetypal decision-making strategies based on $\mu_{F_1}$ values chosen by force-$F_1$.**

| strategy label | $\mu_{F_1}$ | $\phi_{F_1 F_2}$ | $\phi_{F_1} G$ |
|---|---|---|---|
| population strategy | $\left(-\frac{1}{6}, 0\right)$ | $\left(-\frac{\pi}{6}, 0\right)$ | $\left(-\frac{\pi}{2}, -\frac{\pi}{3}\right)$ |
| mixed strategy | $\left(0, \frac{1}{3}\right)$ | $\left(0, \frac{\pi}{3}\right)$ | $\left(-\frac{\pi}{3}, 0\right)$ |
| adversarial strategy | $\left(\frac{1}{3}, \frac{1}{2}\right)$ | $\left(\frac{\pi}{3}, \frac{\pi}{2}\right)$ | $\left(0, \frac{\pi}{6}\right)$ |

## Assumptions and limitations

The model defined in this work presents assumptions of population motivation and behaviour that is not universal for all non-combatants in the vicinity of conflict. For instance, [47, 48] assumed that the population could enter as combatants, and [58] assumed a static population. The inclusion of population initiative terms in Eq (18) introduces a tension between the support a force can generate, and the organisational effectiveness, via Eq (16), they can maintain. Both terms affect combat effectiveness in the Lanchester engagement in Eq (7); changes in organisational effectiveness are felt immediately in each force's lethality, whereas population sentiment changes are felt indirectly through the relative sizes of each supporting population, and their effect on the modulation terms $f$. Though the influence of population sentiment on combat effectiveness is indirect, the combination of both supporting (denominator) and detracting (numerator) population sizes in $f$ means that it can influence combat effectiveness more than the organisational effectiveness of each force. Such assumptions are appropriate where the forces and population have little concern about lethality escalation (such as the British army and Spanish guerilla fighters against the Napoleonic French [10]), and inappropriate when either force wishes to maintain a low-level conflict which does not escalate (such as between Ukrainian forces and pro-Russian separatists/Russian forces in the Donbas region during 2014 [59]). Thus, the utility of this model lies in the ability to articulate its assumptions and explore their consequences, not in any claims of universality of those assumptions.

As detailed in [60], the chosen resolution of a combat model, and its assumptions/limitations, can be used to inform its applicability. For instance, the presence of Lanchester recruitment terms in [5, 45, 46] enables such models in their full generality to be applicable over campaigns which include multiple engagements and requirements to replenish forces. The absence of recruitment terms in Eq (2) for each force means that the model in this work is relevant over a shorter time-frame. The static nature of the networks in Fig 2 and the decision-making strategies in Eqs (20) and (21) also reinforce that outputs from this work are relevant over a single engagement, as these inputs would likely change for a force suffering considerable losses over an extended period of time. We plan to explore the effects of decision-making strategies on combat outcomes in the vicinity of non-combatants in a campaign setting over multiple engagements in future iterations of this work. Moreover, the simplistic use-case with static graphs we examine below allows a baseline understanding of the model before adding further layers of complexity such as dynamical networks.

## Dimensional reduction of decision-making

To reduce the dimensions of the decision-making component of Eq (1), we assume that the phases of each of the three networks have approximately phase synchronised, each network centred on a global phase given by Eq (13). The procedure, which relies on the eigenspectrum of matrices $\mathcal{B}$, $\mathcal{G}$ and $\mathcal{R}$, is a generalisation of Section 2.3 and Appendix B of [42]—we omit the details from here for brevity. Ultimately, the average phase differences $\Delta_{BG}$ and $\Delta_{GR}$ are given by

$$
\begin{aligned}
\dot{\Delta}_{BG} = \bar{\omega}^B - \bar{\omega}^G - \psi_G^B \sin(\Delta_{BG} - \phi_{BG}) - \psi_B^G \sin\Delta_{BG} \\
- \psi_R^B A(B) \sin(\Delta_{BG} + \Delta_{GR} - \phi_{BR}) + \psi_R^G \sin\Delta_{GR},
\end{aligned}
\tag{23}
$$

$$
\begin{aligned}
\dot{\Delta}_{GR} = \bar{\omega}^G - \bar{\omega}^R + \psi_B^G \sin\Delta_{BG} - \psi_B^R A(R) \sin(\Delta_{BG} + \Delta_{GR} + \phi_{RB}) \\
- \psi_R^G \sin\Delta_{GR} - \psi_G^R \sin(\Delta_{GR} + \phi_{RG}),
\end{aligned}
\tag{24}
$$

with the 'renormalised' couplings given by

$$\psi_Y^X \equiv \frac{\zeta_{XY}|\mathcal{I}^{XY}|}{|\mathcal{X}|} \quad \text{for networks } \mathcal{X} \text{ and } \mathcal{Y}, \tag{25}$$

and the mean of each network's natural frequencies defined by

$$\bar{\omega}^X \equiv \frac{1}{|\mathcal{X}|} \sum_{i \in \mathcal{X}} \omega_i^X. \tag{26}$$

Finally $|\mathcal{I}^{XY}|$ is the total number of edges between the networks $\mathcal{X}$ and $\mathcal{Y}$. The force engagement and population model components simplify by each decision-making network assumed to experience high rates of synchrony, *i.e.*

$$O_B(t) \approx O_G(t) \approx O_R(t) \approx 1. \tag{27}$$

Existence and uniqueness of model solutions can be guaranteed; since all of the Kuramoto-Sakaguchi, Lanchester and Lotka-Volterra components, and their derivatives (including simplifications via Eqs (23)–(27)), are real valued and smooth on the intervals

$$\{B \in [0, B(0)], R \in [0, R(0)], g_B \in [0, G], g_R \in [0, G], \theta_i \in \mathbb{S}^1 \ \forall i \in \mathcal{B}, \mathcal{G}, \mathcal{R}\}, \tag{28}$$

the Picard–Lindelöf theorem ensures uniqueness of solutions of any model instance with initial conditions derived from Eq (28), and satisfying consistency condition Eq (11).

## Use-case networks and parameter values

The specific network topologies and natural frequencies are shown in Fig 2. Coloured nodes with common shapes and labels in Fig 2 share links with other networks. Thus, green circles labeled 1–5 in the Blue network share edges with blue circles in the Green network, from which it follows that $|\mathcal{I}^{BG}| = 5$, and $|\mathcal{I}^{BR}| = |\mathcal{I}^{GR}| = 16$. The bottom row of Fig 2 displays each labeled node's internal degree (left) and natural frequency (right). Thus, although Blue shares fewer edges with Green than with Red, the bottom-left panel shows that the 5 most-connected nodes in Blue are connected with the five most-connected nodes in Green. Additionally, we provide a summary of common parameter values and initial conditions used throughout this work in Table 4. These choices reflect two aspects: Blue, Red and Green are able to achieve near synchronisation apart from deviations through instabilities—which shall be explored—and the decision making time-scales are significantly faster than those for the combat dynamics, reflecting an intuitive property that many decisions would be made through any finite period of adversarial engagement.

The Kuramoto-Sakaguchi model under the parameter and initial condition values given in Table 4, and the topology offered in Fig 2, has been studied in our previous work [42], leading

**Table 4. Table of common parameter values and initial conditions used in this work.**

| parameter | value | initial condition | value |
|---|---|---|---|
| $\{\bar{\omega}^B, \bar{\omega}^G, \bar{\omega}^R\}$ | $\{0.503, 0.5, 0.551\}$ | $\{B(0), R(0)\}$ | 2100 |
| $\{\sigma_B, \sigma_G, \sigma_R\}$ | $\{1.5, 0.2, 0.5\}$ | $\{g_B(0), g_R(0), \Gamma(0)\}$ | $\{10^4, 10^4, 4 \times 10^4\}$ |
| $\{\zeta_{BG}, \zeta_{GB}, \zeta_{GR}, \zeta_{RG}\}$ | 0.2 | $\{\Delta_{BG}(0), \Delta_{GR}(0)\}$ | 0 |
| $\{\zeta_{BR}, \zeta_{RB}\}$ | 0.4 | | |
| $\{\kappa_{BR}, \kappa_{RB}\}$ | $10^{-4}$ | | |
| $\{\kappa_{GB}, \kappa_{GR}\}$ | $2 \times 10^{-11}$ | | |

to an understanding of expected behaviours which shall inform the current work. Specifically, careful attention was given to the changes in parameter values which resulted in oscillator equilibrium, or limit cycle behaviour. Mathematical properties of the current model, such as fixed points and potential bifurcations, are detailed in Appendix A in S1 File.

Further addressing the parameters chosen in this work, the force initial conditions and the physical effectiveness parameter values were chosen to represent an engagement between *near-peer* adversaries of equal technological capability. Finally, the population initial conditions and willingness/inertia values were chosen to represent a population significantly larger than either force, both forces have an equal number of supporters, and any change in support happens at a slower time scale than both combat outcomes, and decisions.

As detailed in [61], models are an abstraction of reality. High fidelity combat models which are focused on tactical engagements, such as COMBATXXI [62], are similar to high fidelity models found in biology [63] and chemistry [64], in that they are based on well established physical laws, and can therefore receive a high degree of validation—typically trusted for their predictive abilities [65]. The model detailed in this work is more abstract, concerned with the impact that strategic decisions have on combat outcomes. As such, the underlying theory, and corresponding outputs, are harder to validate—documented challenges include event-validation of the Lanchester model to emulate historical battle data [38], and cross-validation of the Kuramoto model against qualitative decision-making data [22]. Though lacking precise predictive abilities, abstract combat models such as [32, 66] and the current work have much to offer, as they provide a quantitative and dynamic means to understand complex scenarios for which very little data exists. In the next section we highlight how model results can be interpreted in light of events of the 2003 Iraq war.

## Examples of model behaviour

The full system is solved numerically using the 4(5)-order Runge-Kutta-Fehlberg method. Fig 3 offers an example of complex model behaviour when both Blue and Red forces pursue an exclusively adversarial strategy ($\mu_B = \mu_R = 1/2$), namely intending to stay maximally ahead of the adversary's decision-making ($\phi_{BR} = \phi_{RB} = \pi/2$). The top panel shows Blue and Red nearly match each other, but eventually Red sees a rapid collapse with Blue winning the engagement. There are essentially three stages to the engagement, with Blue first ahead up to $t \approx 4000$, then suffering a partial collapse but stabilising at $t \approx 5500$, and finally Red rapidly destroyed from $t \approx 9000$.

Examining the behaviour of some of the underlying factors we begin to unveil the cause of the outcome. In the third panel we see positive values for both $\sin\Delta_{BG}$ and $\sin\Delta_{RG}$ for $t < 5500$. The non-combatants continue to be offered no initiative so they rescind support to both sides. This is reflected in the population proportions in the second panel where neutrals (green curve) come to dominate by $t = 2000$. The inset shows however that this slightly changes later in the dynamics, which will be addressed. Within the window $t < 5500$, the third panel shows that $\sin\Delta_{BR}$ has the largest value, thus Blue maintains a persistent decision advantage over Red and enjoys superior force strength through this stage.

At $t \approx 4000$, the continuous loss of non-combatant support sufficiently degrades Blue's strength to weaken its interaction with Red in the decision dynamics; this causes $\Delta_{BR}$ to slowly change sign, becoming negative at $t \approx 5000$ (third panel of Fig 3, the blue-dash-red-dot line). The transition from equilibrium to dynamic behaviour of the Kuramoto-Sakaguchi component can be understood as the change in force values affecting the system stability—refer to Appendix A in S1 File for more details. This change negatively impacts Blue's effectiveness against Red, as seen in the $B - R$ plot. Additionally, Blue's relaxation from Red sees Green gain

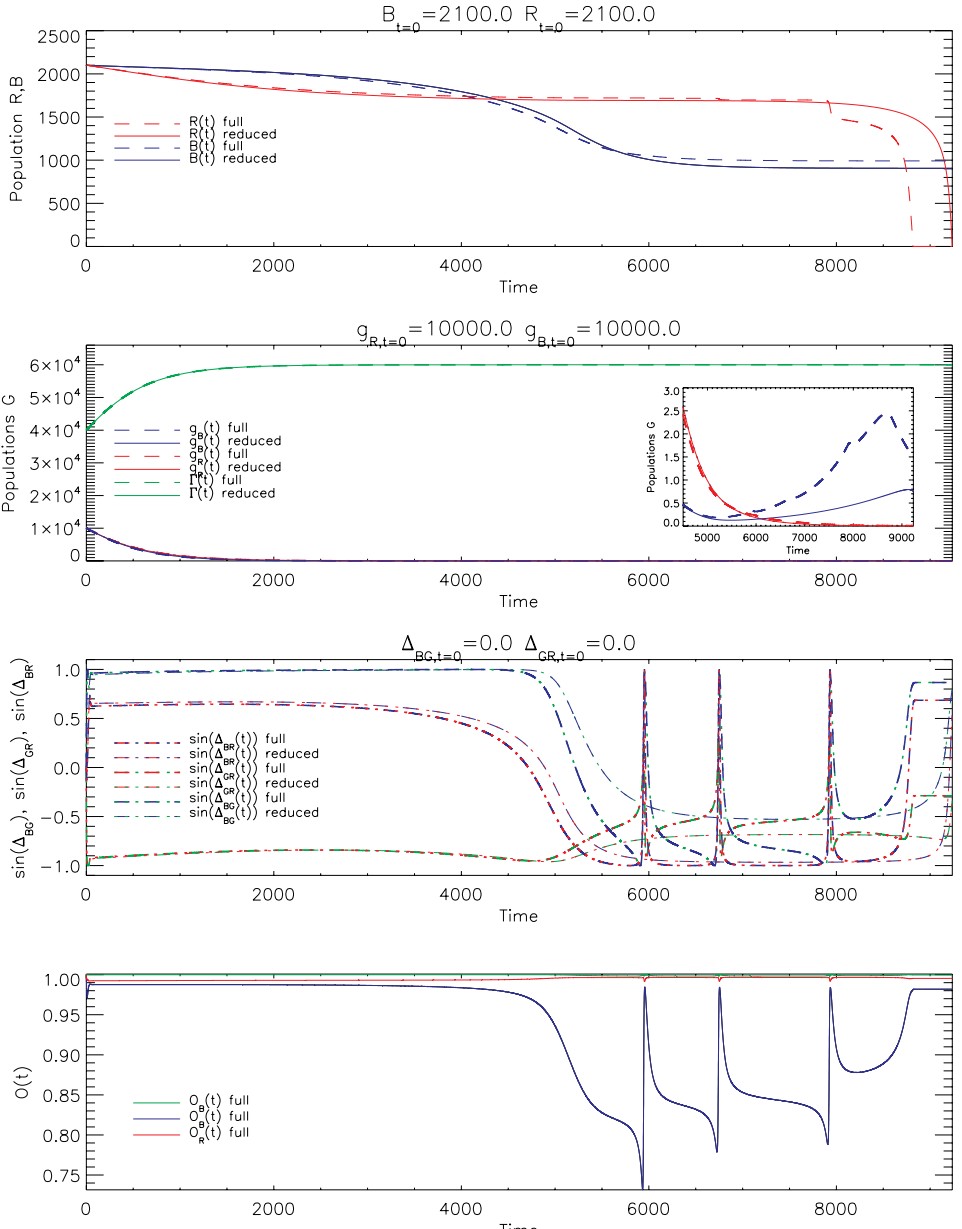

**Fig 3. System behaviour of the reduced system (solid curves) with the full system (dashed curves) for $\mu_B = \mu_R = 1/2$, leading to $\phi_{BR} = \phi_{RB} = \pi/2$, and $\phi_{BG} = \phi_{RG} = \pi/6$.** The inset in the second plot, for population values, shows a small rise in Blue supporters which is not noticeable on the main plot. The bottom plot shows the order parameters obtained from the full system.

initiative ahead of Blue shortly after—seen in the third panel, where the blue-dash-green-dot curve begins to drop and becomes negative at $t \approx 5500$, also leading to fluctuations between the different groups rotating with respect to each other through $6000 \geq t \geq 8000$. When the dynamics stabilises, Blue continues to cede initiative to the population, with $\sin\Delta_{BG} < 0$ for $t > 8000$. This coincides with Red's dramatic defeat, with Blue returning to its original strategy with respect to Green.

The inset in the second panel shows population support for Blue growing very slightly from $t = 6000$ for the first time in the engagement. This support is sufficient for Blue to match Red's

success at being ahead of Blue's decision-making cycles. Further beyond this point, Red continues to stay ahead of Blue's decision-making, with both force numbers remaining approximately static. Nevertheless, at $t \approx 8000$, population support for Blue begins to over-match the benefit Red receives from staying ahead of Blue's decision cycles, beyond this point, culminating in Red defeat.

The trajectories presented in Fig 3 align relatively well with the US/coalition perspective of the first years of the 2003 Iraq war [67]. Specifically, the start of the war in 2003 saw a relatively quick collapse of the Saddam Hussein regime, followed by a number of years of protracted and ambiguous conflict with the emerging insurgency due to the coalition failing to prevent looting after the regime's collapse, in combination with the de-Ba'athification of senior Iraqi positions being interpreted as the trajectories presented between $0 < t < 5000$ in Fig 3. The Patraeus doctrine was introduced in 2007 which saw both the Surge [68]—in order to enhance security afforded to non-combatants and increase internal governance—and US forces recruiting Sunni tribe members, helping diminish insurgent violence in the second half of 2007. These actions can be interpreted as coalition forces affording initiative to the population, thus resulting in the change in $\Delta_{BG} < 0$ for $t > 5000$—leading to population support for Blue.

The bottom panel of Fig 3 gives the synchronisation of each network. Notably, the loss of intensity of interaction from Blue to Red negatively affects Blue's synchronisation. The fluctuations in $O_B$ from $t = 5500$ in that panel match the dynamics seen in those in the third panel. However, the order parameter recovers towards the end, playing a role, along with its support from Green, in the rapid collapse of Red strength.

Fig 3 generally displays quite good agreement between the full and reduced models; largely explained by the relatively large synchronisation levels for each of the networks, which is the basis for the approximation offered by Eqs (23) and (24). Nevertheless, we can see where disagreement between the two solution methods manifests, especially with regards to the trajectory of Red for $t > 8000$. Additionally, $\sin\Delta$ values for the full system perform four brief revolutions across $\mathbb{S}^1$ for $t > 6000$, in contrast to the corresponding values for the reduced system. This disagreement can be understood by noticing deviations of the Blue's synchronisation values in the bottom right panel of Fig 3 for $5000 < t < 8000$. Though the validity of Eqs (23) and (24) is challenged by these deviations, the impact is limited, since order parameter values of $O > 0.7$ do nonetheless reflect a moderate degree of synchronisation.

## Impact of decision-making strategies

The remainder of this work examines the effect of force strategy choices on outcomes, displaying how model behaviours may be interpreted in terms of recognisable features of warfare through the lens of mathematical constructs. Fig 4 provides force strengths and populations for the full computational model at the final time $T_{end}$, when either $B$ or $R$ first reaches zero value. Thus the top left panel presents the difference in final force strengths $B - R$ at the end of engagement. Colour blue or red corresponds to which of Blue or Red wins; white represents stalemate attrition. The middle left shows the number of Blue supporters in Green, middle right the Red supporters, and top right the neutrals. The bottom left of Fig 4 presents the final time, $T_{end}$. The bottom right shows the time-average of Blue's order parameter

$$\langle O_B \rangle = \frac{\int_0^{T_{end}} dt O_B(t)}{T_{end}}. \tag{29}$$

Each panel is a function of the strategy choice of Blue (horizontal, $\mu_B$) and Red (vertical ($\mu_R$). We superimpose on each panel a division into nine sections (the dashed lines),

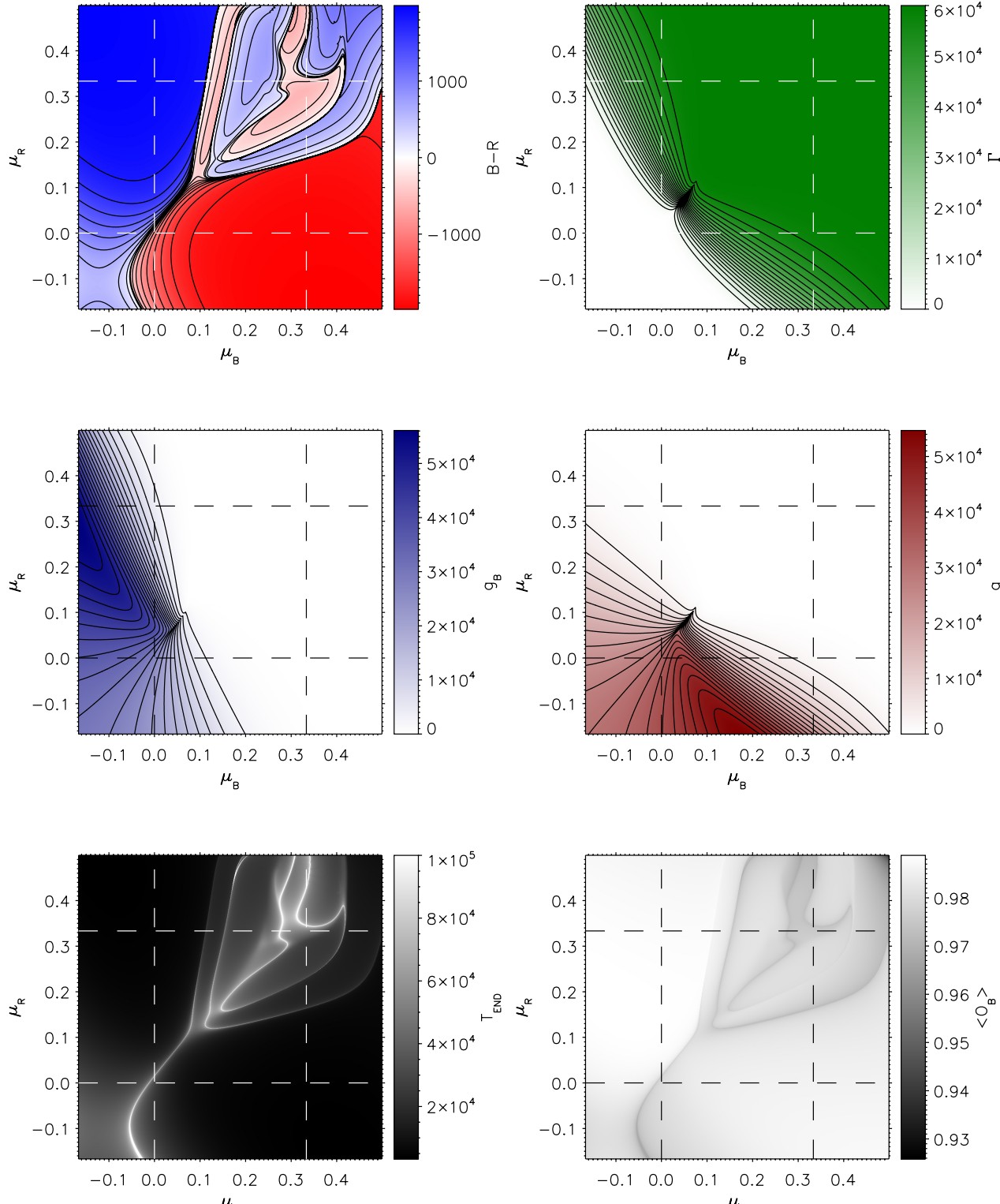

**Fig 4. Final outputs of the full system while varying the decision-making strategies $\mu_B$ (horizontal axes) and $\mu_R$ (vertical axes).** Each panel consists of $960 \times 960$ end states over the parameters $\mu_B$ and $\mu_R$. Top row presents $B - R$ (left) and $\Gamma$ (right), middle row presents $g_B$ (left) and $g_R$ (right), and the bottom row presents final time $T_{end}$ (left) and average order parameter value $\langle O_B \rangle$ (right) corresponding to the time taken for either Blue or Red forces to reach zero value.

corresponding to the archetypal strategy pursued by each force—population, mixed or adversarial—as given in Table 3.

The top-left panel of Fig 4 displays continuously connected regions of Blue or Red victory, boundary regions of stalemate, and regions of significantly variable outcome with abrupt shifts between Blue and Red victory with small parameter variations. Similar outcome representations were employed in [35], deliberately constructed akin to a *phase space* in thermodynamics, distinct from the phase of an oscillator. These phase boundaries in $B - R$ also match the structures seen in $T_{end}$ (top right).

Three types of geometrical constructs can be identified in the phase plot. A single continuous line from bottom left to nearly the middle divides two clear phases corresponding to Blue (above) or Red (below) victory. This line ends at a 'critical' point at approximately (0.1, 0.1). To the top right of the point, the more abrupt complex transitions can be found, bounded above and below by separating (from the critical point) continuous lines; the example given in Fig 3 is a point in the centre of this region.

Forces which intend population strategies ($-1/6 < \mu < 0$) tend to fare better than those which pursue mixed or adversarial strategies. Indeed, if either force chooses a population strategy, and their adversary does not, that force is assured victory—except in a small region at the boundary of $\mu_B = \mu_R = 0$, where Red achieves victory while pursuing a mixed strategy. Model dynamics dictates that the phase boundary line lies outside the joint population-strategy region, and the critical point sits inside the jointly mixed strategy region.

We can understand that victory favours population strategies as a consequence of the relevant supporting populations growing in size, as seen in the bottom left and middle panels of Fig 4 for $g_B$ and $g_R$ values, respectively. A large supporting population enables the modulation feedback $f$ in Eq (7) to grant superior combat effectiveness to the relevant force, in addition to hindering the adversary—this behaviour is explicitly built into the model. Nevertheless, there are evident asymmetries in the contours of Fig 4. For example, Blue is generally favoured when both forces pursue a pure population, and pure adversary strategy, as seen in the bottom left and top right corners of the $B - R$ outcome. The mechanisms behind this asymmetry for population strategies shall be explored shortly using the reduced model.

When both forces pursue a mixed strategy, or one force pursues a mixed and the other an adversarial strategy, there is the third type of geometric structure, a phase boundary where on one side one force dominates the engagement, and on the other side the dominance disappears, leading to the other force obtaining victory, but at low remaining force values. This lies inside the region of sharp changes in outcome with small parameter variations, and is where further asymmetries are evident. Focusing on $\mu_B \approx 0.1$ and $\mu_R \gtrsim 0.1$, to the left Blue dominates, and to the right Red obtains victory, albeit with low surviving force strength values. In the Blue dominant region, the population supports neither force—seen in the green dominance of the top right panel of Fig 4, because no decision-making initiative is afforded to the population. However, this absence of support by Green occurs such that the modulation term for Blue is small in contrast to that for Red for the majority of the time, thus enabling Blue's decisive victory. On the other side of the boundary, the engagement begins much the same, however, the rescinding of population support reaches a tipping point, where the non-combatant population numbers stabilise before Red reaches zero force strength. Thus population support for Red is now comparatively higher and Red force strength stabilises, albeit at low values. The Blue force experiences a crash similar to that for Red in Fig 3.

Finally, the bottom-right plot shows the the average Blue order parameter, as its hierarchical topology lends itself to being the least synchronised of the three networks. Generally the order parameter obtains very high values, $\langle O_B \rangle > 0.97$ (more white), outside the complex region of the $B - R$ heatmap. On the phase boundaries, the value decreases, but never below 0.93. This

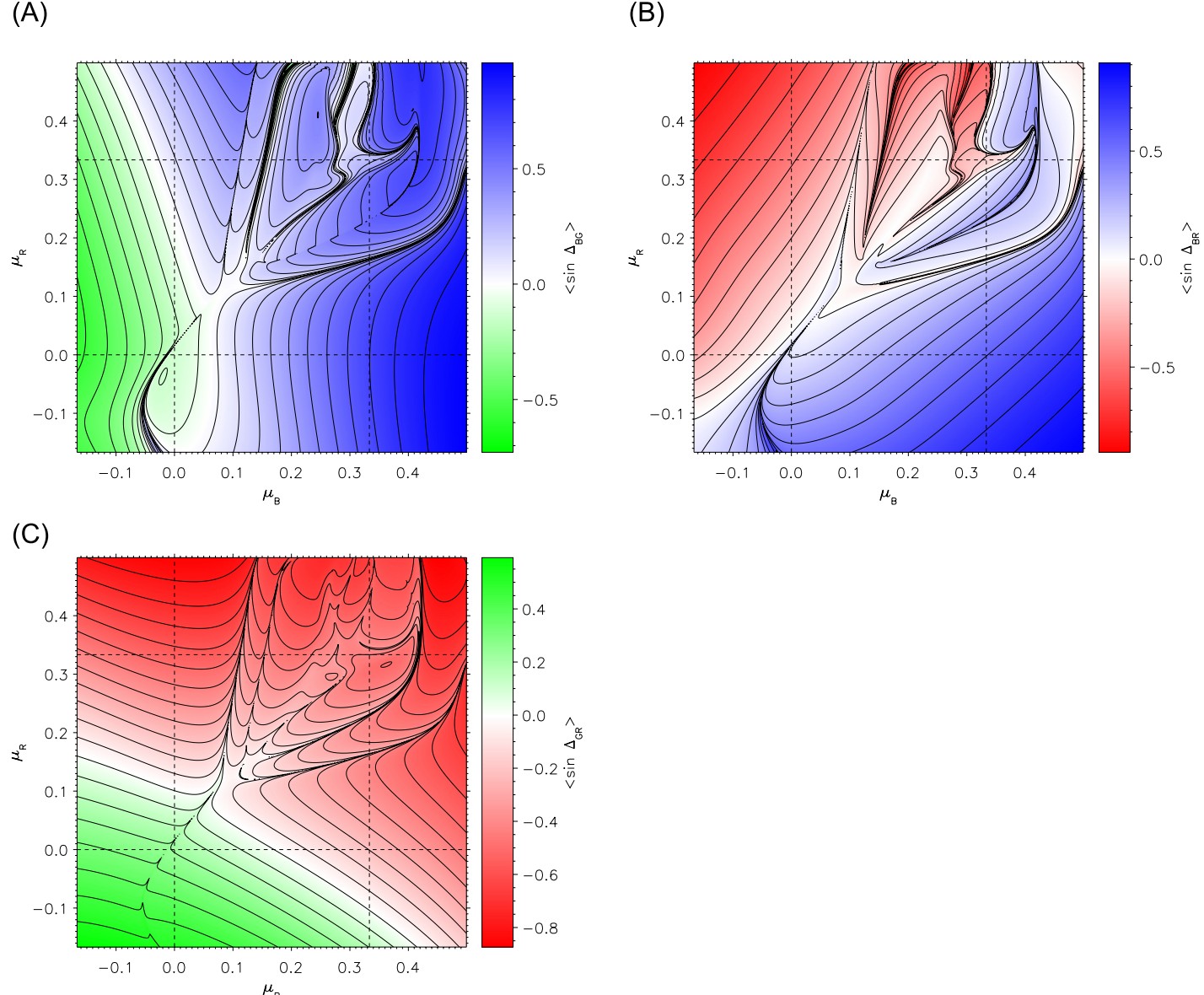

**Fig 5. Temporal averages of phase difference values, $\langle \sin\Delta_{BG} \rangle$ (left), $\langle \sin\Delta_{BR} \rangle$ (right), and $\langle \sin\Delta_{GR} \rangle$ (bottom), while varying the decision-making strategies $\mu_B$ (horizontal axes) and $\mu_R$ (vertical axes).** Similar to Fig 4, each panel consists of $960 \times 960$ end states over the parameters $\mu_B$ and $\mu_R$.

decrease can be understood by the longer engagements temporarily straining Blue's synchrony, as witnessed in the bottom right panel of Fig 3. These boundaries correlate well with the engagement time $T_{end}$, signifying drawn-out attritional stalemates. Nevertheless, consistently high order parameter values are seen in the majority of the scanned parameter space, leading to the expectation that Eqs (23)–(27) are valid away from the phase boundaries.

Fig 5 presents the time-averaged values of phase differences, $\langle \sin\Delta \rangle$, given by

$$\langle \sin\Delta \rangle = \frac{\int_0^{T_{end}} dt \, \sin\Delta(t)}{\int_0^{T_{end}} dt}. \tag{30}$$

Some structures are replicated from Fig 4, particularly a portion of the critical line at the bottom left in all three panels, the critical point at (0.1, 0.1), and the complex region in the upper right. However, there are new gradual transitions in both $\langle\sin\Delta_{BG}\rangle$ and $\langle\sin\Delta_{GR}\rangle$, where green transitions through white to blue or red. These are not reflected in $B - R$, but do correlate quite well with the final values of $g_B$ and $g_R$, given in the middle panels of Fig 4. This is intuitively sensible as the expressions $\sin\Delta_{GB}$ and $\sin\Delta_{GR}$ in the population initiative term in Eq (18) determine the growth/decline of population support for either force.

Discussing the top right panel of Fig 5, recall the example of Fig 3 that sits in the complex region of the heatmap plots. As described, Blue dynamically adjusted its strategy in relation to Green, allowing for initiative, then returning to its intended adversarial strategy after Red's defeat. The regions of blue or red in the top right panel of Fig 5 are generally anti-correlated with the outcomes in $B - R$, concurrently with Blue or Red achieving success by permitting Green initiative (as seen in left and right panels).

There are exceptions to this, such as the complex upper-right regions where the outcome of $B - R$ generally correlates with the sign of $\Delta_{BR}$. These are therefore points in strategy space where the Boyd maxim of decision advantage against the adversary leads to tactical success. It follows, nonetheless, from the dynamics of the nonlinear system, not necessarily the intended decision strategy. Nevertheless, in this complex system of divided population, the advantage generally comes from allowing initiative in supporters at the expense of seeking it over the adversary. An 'anti-Boyd' approach may be seen to be valid in large portions of the phase space. Fig 6 shows how the decision advantage factor correlates with outcome. Pure correlation would see shaded regions in the bottom-left and top-right, while pure anti-correlation would reverse this. Fig 6 displays regions of anti-correlation: increasingly dense regions are seen to

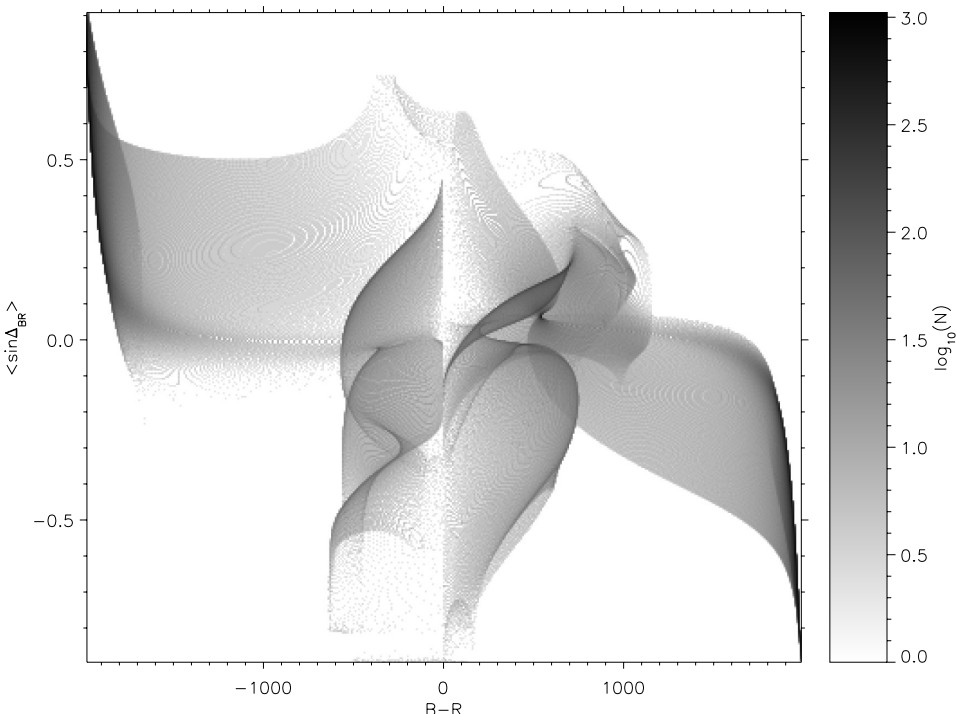

**Fig 6. Plot of final $B - R$ values (horizontal axes) against final $\langle\sin\Delta_{BR}\rangle$ values (vertical axes) for all the $960 \times 960$ end states used to produce Figs 4 and 5 over the parameters $\mu_B$ and $\mu_R$.** Darker colours signify a greater number of final $B - R$ and $\langle\sin\Delta_{BR}\rangle$ values occurring simultaneously.

coincide with larger victory margins. Conversely, narrower margins (light-grey to white) correlate with seeking an advantage over the adversary. This underscores the insight from this model that supporter initiative—and not just pure advantage against the adversary—is an important factor in a multi-party conflict.

## Comparison with reduced system

Fig 7 displays the difference between the final outputs of the full computational system, and the final outputs of the reduced system given by Eqs (23)–(27). Deviations between the two approaches occur on the boundaries, where the reduced model is unable to immediately account for the change in behaviour and outcomes as $\mu_B$ and $\mu_R$ vary. Given that the average order parameter value $\langle O_B \rangle$ differs from unity more strongly on these boundaries (bottom-right panel of Fig 4), the deviation of model outcomes between the two approaches is not unexpected in these regions. Nevertheless, the strong correlation between these two approaches on either side of the boundaries confirms that the reduced system offers a computationally inexpensive means to analyse system behaviour.

Given the success of the reduced system away from the contours, we employ it to explore why Blue is favoured when both forces pursue a pure population decision-making strategy. In this case, the main parametric differences of each force involve the mean value of each network's natural frequency distribution, and how many edges are shared between Green and each force. Additionally, as evident in the middle panel of Fig 5, the victorious force (almost exclusively Blue) in the region of pure-population strategy generally obtains superior levels of organisational effectiveness (*i.e.* $\Delta_{BR} > 0$), except in the upper-left corner of the population-strategy region around the point $\{\mu_B, \mu_R\} = \{-1/6, 0\}$ which go against this trend. Thus,

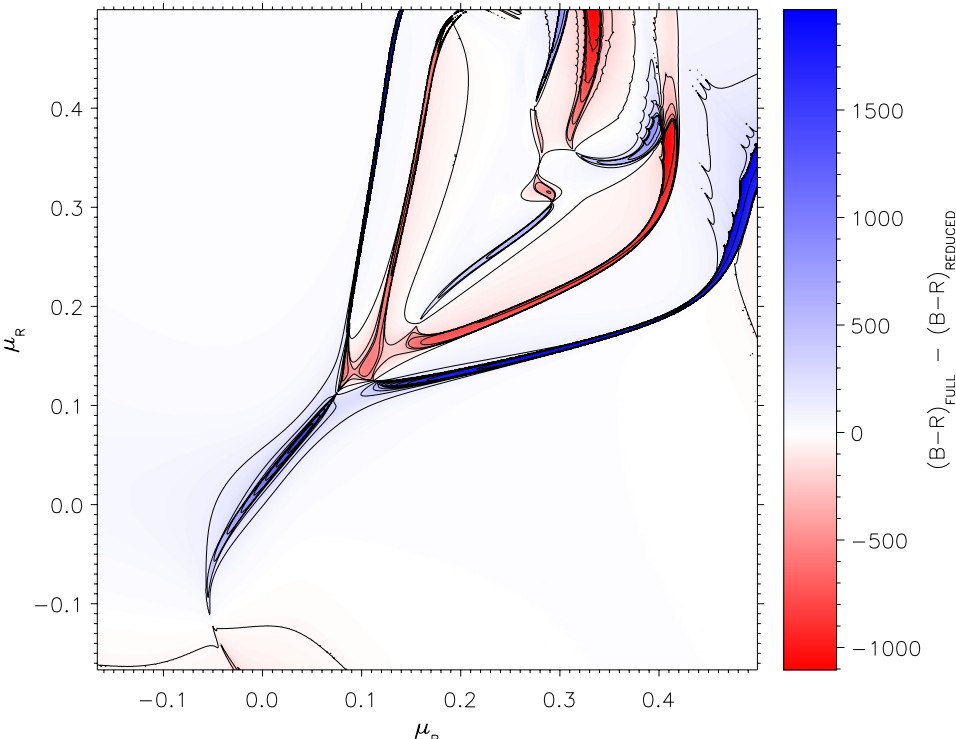

**Fig 7. Difference between the final outputs of the full computational system, and the final outputs of the reduced system Eqs (23)–(27), while varying the decision-making strategies $\mu_B$ (horizontal axes) and $\mu_R$ (vertical axes).**

focusing on the values $\{\mu_B, \mu_R\} = \{-1/12, -1/12\}$, which avoid the region where the trend is not followed, we are guided by the observed phenomena

$$
\begin{aligned}
\Delta_{BR} > 0, \quad &\text{leads to Blue victory} \\
\Delta_{BR} < 0, \quad &\text{leads to Red victory.}
\end{aligned} \tag{31}
$$

It is possible to obtain an approximate expression for the initiative between Red and Blue, given by:

$$
\Delta_{BR}(B, R) \approx 2\tan^{-1}\left(\frac{C(B,R) - \sqrt{C^2(B,R) + S^2(B,R) - \chi^2}}{\chi - S(B,R)}\right), \tag{32}
$$

where $C \equiv C(B, R)$, $S \equiv S(B, R)$, and

$$
C = \psi_R^B\left(\frac{B}{B_0}\cos\phi_{BR} + \frac{R}{R_0}\cos\phi_{RB}\right), \quad S = \psi_R^B\left(\frac{B}{B_0}\sin\phi_{BR} - \frac{R}{R_0}\sin\phi_{RB}\right), \tag{33}
$$

$$
\chi = \bar{\omega}^B - \bar{\omega}^R - [\psi_G^B\sin(\Delta_{BG}^* - \phi_{BG}) + \psi_R^G\sin(\Delta_{GR}^* + \phi_{RG})]. \tag{34}
$$

The derivation of Eq (32) is given in Appendix B in S1 File. Notably, Eq (32) assumes that the force values $R$ and $B$ change on a slower time scale than the phase differences $\Delta$ (individual decisions are faster than the pace of conflict), and that the system given by Eqs (23) and (24) for static force values reaches equilibrium, as opposed to a limit cycle. We employ Eq (32) to approximate the contour when $\Delta_{BR} = 0$, given by

$$
\Delta_{BR}(B_0, R_0) = 0 \;\Rightarrow\; S(B_0, R_0) = -\chi \tag{35}
$$

where either side of the contour given by Eq (35) can be used to test the assumptions laid out in Eq (31). Notably, Eq (35) assumes static force values, fixed at $\{B, R\} = \{B_0, R_0\}$.

Thus, deviating away from the use-case frequency values and topology of Figs 2 and 8 (left) displays the final outcome for the reduced system of each force varying its average $\omega$ value, and (right) shared edges with Green $|\mathcal{I}_{BG}|$ and $|\mathcal{I}_{GR}|$. Again, both forces pursue equivalent population decision-making strategies of $\mu_B = \mu_R = -1/12$. The use case parameter values are denoted by black diamonds on both panels. The left panel shows both steady state and limit cycle regions, the latter highlighted by the black hatching. The contour patterns demonstrate that the outcome is sensitive to the average natural frequency values of each force. The analytically determined contour of Eq (35), shown in white in Fig 8, provides some predictive power in determining the final outcome when the system given by Eqs (23) and (24) demonstrates approximately steady state behaviour. In particular, along this line the two sides (consistent with the slow dynamics approximation) are gradually undergoing attritional stalemate. This stalemate line also matches the transitions on either side of the contour where the reduced system holds. We postulate that the accuracy of this contour to predict final outcomes could be increased, but the added complexity would no doubt destroy the analytic nature of the approach.

The right panel of Fig 8 clearly demonstrates a dependence of the outcomes on Blue and Red's connectivity with Green. When both sides pursue the same population strategy, the victorious force is the one that cedes *marginally* less initiative to Green than their adversary while nonetheless seeking to retain advantage over the adversary. Examination of individual time-series (omitted here for brevity) reveals the total neutral population becomes negligible almost immediately, with both forces obtaining nearly equal share of population population support. Nevertheless, the successful force invariably provides slightly-less initiative to Green than their adversary, and consequently obtains slightly-less population support. This has a follow-on

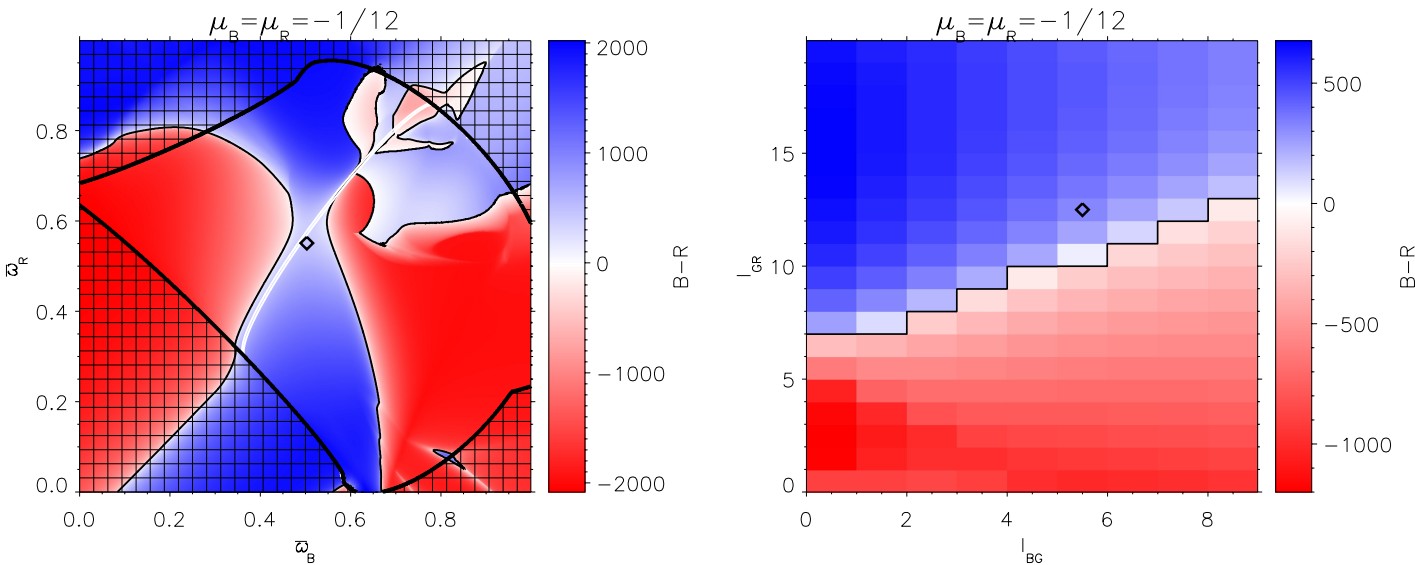

**Fig 8. Effect of changing average ω values (left panel) and edges shared with Green (right panel) of the reduced system when both forces pursue equivalent population decision-making strategies, with the use case values denoted by black diamonds on both panels.** In the left panel the white line shows the contour given by Eq (35), and the hatched areas indicate parameter values resulting in Eqs (23) and (24) exhibiting limit-cycle behaviour for nearly static force values $B = B_0$ and $R = R_0$.

impact on the lethality coefficient—again nearly equal, with marginal differences—and the organisational effectiveness.

Thus, the victorious force on the right hand panel of Fig 8 finds a strategic *sweet-spot*, where enough initiative is afforded to the population to enable meaningful support, whilst also obtaining a decision-making advantage over the adversary, echoing the discussion around Fig 6. Clearly, the use case values on the right hand panel favour Blue, as it has significantly less connection to Green in this case. Being *less responsive* to Green than Red in this case, thanks to overall less connectivity, affords Blue greater organisational effectiveness against the adversary, thus securing victory. Red achieves victory when its number of connections with Green reduces, importantly though Red is able to maintain significantly more connections to Green while still obtaining victory.

One more factor underlies these results, explains the behaviours as numbers of connections to Green by either side reduce to zero, and the overall offset of the otherwise symmetric pattern in the right hand panel of Fig 8. This lies in Red having marginally higher average natural frequency (given the specific instance from a uniform distribution of frequencies). This slight advantage permits Red to maintain more connections with Green than Blue, while not undermining its ability to achieve advantage against Blue, and thus obtain victory.

Although the topology of both decision-making networks are different, the reduced system given by Eqs (23) and (24) has lost such topological information, replaced instead by total number of links via Eq (25). Thus, the only difference between Red and Blue being measured in this case, aside from connectivity with Green, is their average natural frequency values. This suggests that Red's higher value enables it to have greater connectivity with Green, whilst also maintaining greater organisational effectiveness than Blue for comparable levels of population support.

## Impact of combat on boundary contours

Solving for $B − R$ as a function of the decision-making strategy is a formidable analytical challenge that we have not succeeded in. Nevertheless, insight may be gained through themore

soluble reverse path: to understand the boundaries seen in Figs 4–7 by approximating the decision-making and combat components of the model. The approximate form of $\Delta_{BR}$ given in Eq (32), as a function of $B$ and $R$ values, enables appreciation of the relative importance of the specific trajectory in $B$ and $R$ at the critical boundaries in phase space. To study the boundary between Red and Blue victory, we derive $\langle\sin\Delta_{BR}\rangle$ values by approximating $B$ and $R$ paths as linear, and then compare the result with the $\langle\sin\Delta_{BR}\rangle$ values and structures shown in Fig 5. Importantly, $\Delta_{BR}$ in Eq (32) is a function entirely in terms of $B$ and $R$ force values, thus the plot of $\langle\sin\Delta_{BR}\rangle$ in Fig 5 can equivalently be derived in terms of the following line integral formulation [69]

$$\langle\beta(\Delta)\rangle = \frac{\int_{\mathcal{C}} ds\,\beta(\Delta(B,R))}{\int_{\mathcal{C}} ds} \quad \text{where} \quad ds = \sqrt{\dot{B}^2 + \dot{R}^2}\,dt \tag{36}$$

for general function $\beta$, and $\mathcal{C}$ is the specific trajectory in the $B$ and $R$ plane corresponding to the values of $\mu_B$ and $\mu_R$. To apply Eq (36), each trajectory is approximated via linear paths. This requires using the final value of the engagement (as given in the top-left panel in Fig 4)—details of this procedure are given in Appendix C in S1 File.

The left hand panel of Fig 9 gives the resulting values of $\langle\sin\Delta_{BR}\rangle$ obtained from assuming $B$ and $R$ follow linear trajectories. Notably, regions where the steady-state assumption used to derive Eq (32) fail are concentrated in the top-right corner, indicated by the hatched area. For more detail about Kuramoto-Sakaguchi fixed-point behaviour, refer to Appendix A in S1 File. Comparison with the top right panel of Fig 5 (reproduced in the right hand panel of Fig 9) shows that values generated from the analytic form of $\langle\sin\Delta_{BR}\rangle$ match quite well with those generated numerically; most of the phase boundary structure well into the complex region of the heatmap is reproduced. Specifically, the critical point at $(0.1, 0.1)$, the outer boundary of the complex region in the top right hand corner and even some of its internal structure match. The approximation is also successful in reproducing the outer layers inside that boundary. Where the approximation breaks down is in reproducing the details of the actual sign of $\Delta_{BR}$ close to the diagonal, even though the shape of the contours is suggestive.

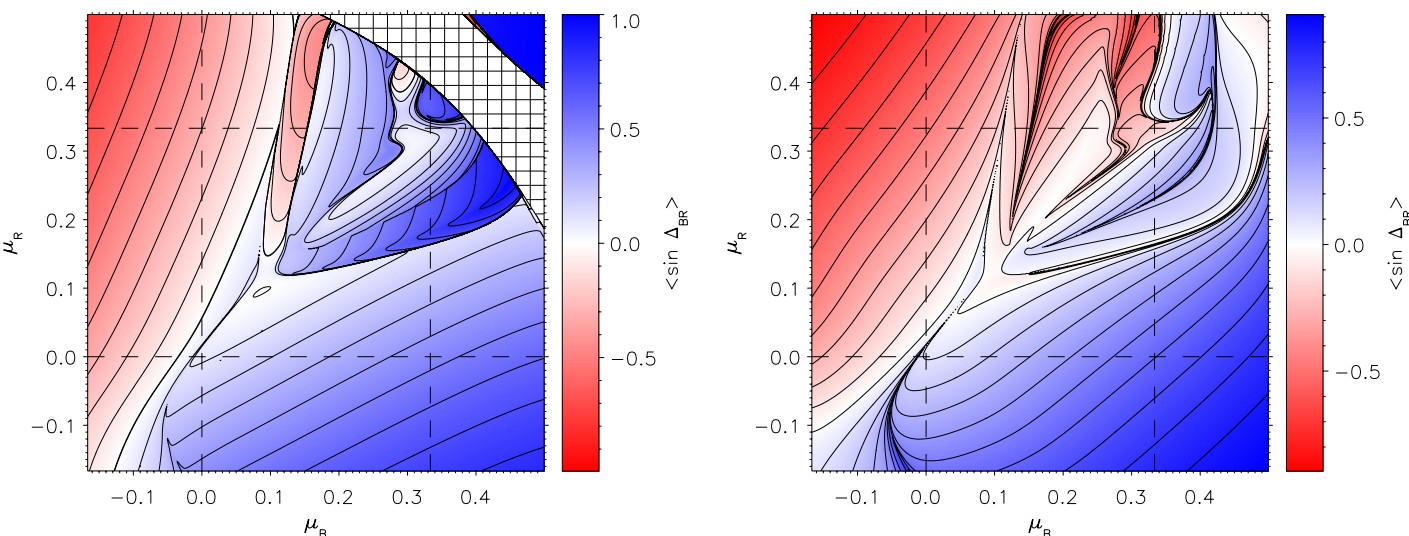

**Fig 9. The left-hand plot shows contours of approximated $\langle\sin\Delta_{BR}\rangle$, given explicitly in Equation (62) of Appendix C in S1 File.** Regions where the steady-state assumption of Eq (32) are not valid are presented as a hatched area, explained in more detail in the Appendix C in S1 File. The right-hand plot reproduces the fully numerical $\langle\sin\Delta_{BR}\rangle$, reproduced from the top right panel of Fig 5.

Fig 9 demonstrates that knowledge of the final outcome between $B$ and $R$ enables a semi-analytic understanding of the decision-making dynamics of the model as a function of $\mu_B$ and $\mu_R$—up to a point. The further $\mu_B$ and $\mu_R$ deviate from population strategies, the less well the linear approximation of the $B$ and $R$ trajectories work to understand the corresponding decision-making dynamics. Evidently, adversarial strategies involve complicated dependencies, in contrast to the benign nature of population strategies. To reinforce the point, combat effectiveness for the force here is mainly derived from population support. For this reason, the combat dynamics here can be approximated as linear paths. Contrastingly, the scenario when mixed and adversarial strategies are chosen displays more complex dynamics, challenging analytical approaches.

## Conclusions and future work

We have presented a model representing complex dynamics among multiple parties in a conflict environment. The model reflected the dynamics and interactions of conflict, where two of the parties seek both cooperative (with the population) and competitive (against each other) decision making. This was achieved by combining the Lotka-Volterra, Lanchester and multi-network Kuramoto systems. The resulting formulation reflected that supporters need to be engaged by the adversaries within the conduct of conflict in order to gain advantage. We have solved the model by numerical and dimensional reduction (semi-analytical) means.

The model provided new insights into the trade-offs in decision making between the need for advantage over an adversary and initiative to supporters or partners. The results were visualised as a heatmap or phase space of conflict outcomes, analogous to that used in thermodynamics. Here we found regions where the traditional (two parties, or one-on-one competition) Boyd maxim of "aheadedness in decisions enables victory" was applicable. Indeed, we could identify semi-analytically such regions and understand the outcomes in terms of fine balances between engagement with neutrals versus the adversary, and how other intrinsic factors such as decision-making speeds could play a decisive role.

However, in this complex environment, we also discovered that in large regions of the phase space, in fact, intended strategies that enabled supporter initiative at the cost of adversary advantage, led to superior outcomes. This may appear counter-intuitive, but certainly echoes the guidance offered in counter-insurgency doctrines. However, such manuals rarely provide a tool for exploring these concepts. Quantitative trade-off analysis may be possible, after more substantial validation, by a model such as we have formulated. One result, given somewhat artificial but nonetheless distinct use-cases for the organisational structures of each side, was that a population strategy for one party may deny any region of phase space where the other may succeed. Topologically this followed from the specific structure of interaction between the advantaged side and its potential supporters, namely focusing on the leadership hub.

Our dimensional reduction results showed that for the majority of the phase space of decision strategies, inexpensive computational resources may be applied to derive results. However, critical points/boundaries in the phase space are sensitive to the full non-linear dynamics. Certain approximations, where the conflict outcome is used as inputs into the decision-making dynamics, provide further insight. This analysis further highlights that there are 'benign' regions of phase space, where linear dependencies on outcomes are essentially valid, even at critical points. However, there is a deeply turbulent region, where our efforts at approximation break down. Here, patches in phase space of outcome advantage shift under small variations of decision strategy. Arguably, generalising these results to the real world represents high-reward and high-risk options, where a marginal error in strategy may lead to significantly different results.

Overall, this work shows that the mathematical formulation of a complex system of multiple actors in a conflict environment is possible, and outputs resonate with the experience often

distilled into doctrine and manuals. Naturally, complex environments such as seen in recent conflicts are even more entangled than the simple tripartite division in our model: humanitarian agencies, private corporations, criminal actors and the social medium of the home societies of the external 'Blue' or 'Red' adversaries. This model nevertheless shows how such structure can be mathematically explicated. While point-by-point validation of such a model to the level of enabling predictive power is potentially unrealistic, the model offers a non-trivial tool for military leaders to examine nuanced concepts, such as the policy applied to non-combatant populations. Though the possibility of advantage through population support is an input of the model, it is the interplay of this with other factors, and the ability to explore trade-offs between structure, posture, and the corresponding impact on mission outcomes, that is the main result of this work.

## Supporting information

**S1 File. Mathematical appendices.**
(PDF)

**S2 File. Adjacency matrix.**
(CSV)

**S3 File. Natural frequency values.**
(CSV)

**S1 Fig.**
(EPS)

## Author Contributions

**Conceptualization:** Mathew Zuparic, Sergiy Shelyag, Maia Angelova, Alexander Kalloniatis.

**Formal analysis:** Mathew Zuparic.

**Funding acquisition:** Mathew Zuparic, Maia Angelova, Alexander Kalloniatis.

**Investigation:** Mathew Zuparic, Sergiy Shelyag, Maia Angelova.

**Methodology:** Mathew Zuparic, Sergiy Shelyag, Maia Angelova.

**Project administration:** Mathew Zuparic, Alexander Kalloniatis.

**Resources:** Mathew Zuparic.

**Software:** Sergiy Shelyag, Ye Zhu.

**Supervision:** Mathew Zuparic, Alexander Kalloniatis.

**Validation:** Mathew Zuparic, Sergiy Shelyag, Ye Zhu.

**Visualization:** Mathew Zuparic, Sergiy Shelyag, Maia Angelova, Ye Zhu, Alexander Kalloniatis.

**Writing – original draft:** Mathew Zuparic, Sergiy Shelyag, Alexander Kalloniatis.

**Writing – review & editing:** Mathew Zuparic, Sergiy Shelyag, Alexander Kalloniatis.

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
