## [Decision Letter · Decision Letter 0]

31 Oct 2022

PONE-D-22-26713`Friend or foe' and decision making initiative in complex conflict environmentsPLOS ONE

Dear Dr. Zuparic,

Thank you for submitting your manuscript to PLOS ONE. After careful consideration, we feel that it has merit but does not fully meet PLOS ONE’s publication criteria as it currently stands. Therefore, we invite you to submit a revised version of the manuscript that addresses the points raised during the review process.

We look forward to receiving your revised manuscript.

Kind regards,

Miguel A. F. Sanjuán

Academic Editor

PLOS ONE

Journal Requirements:

"This research was a collaboration between the Commonwealth of Australia (represented by the Defence Science and Technology Group) and Deakin University through a Defence Science Partnerships agreement. This research has been undertaken with the assistance of resources and services from the National Computational Infrastructure (NCI), which is supported by the Australian Government, and from the OzSTAR national facility at Swinburne University of Technology"

"The work reported in this paper was funded by Australia’s Defence Science and Technology Group through the Modelling Complex Warfighting Strategic Reseach Initiative.

Grant numbers are not applicable, internal Department of Defence (Australia) program.

Website: https://www.dst.defence.gov.au/

**Additional Editor Comments:**

As you see, there are two reports, one suggesting a major revision and another one accepting, though the report is suggesting some very minor possible changes. Please go through all of them and prepare a new version of the manuscript.

Reviewers' comments:

Reviewer's Responses to Questions

**Comments to the Author**

1. Is the manuscript technically sound, and do the data support the conclusions?

Reviewer #1: Partly

Reviewer #2: Yes

2. Has the statistical analysis been performed appropriately and rigorously? 

Reviewer #1: N/A

Reviewer #2: N/A

3. Have the authors made all data underlying the findings in their manuscript fully available?

Reviewer #1: Yes

Reviewer #2: Yes

4. Is the manuscript presented in an intelligible fashion and written in standard English?

Reviewer #1: Yes

Reviewer #2: Yes

5. Review Comments to the Author

Reviewer #1: In general, this is an interesting article. However, an in-depth major revision is recommended before this manuscript could be recommended for publication in such an interdisciplinary Journal as PLOS ONE.

The authors do provide an extensive literature overview and carefully introduce the components of their model. However, the discussed model still lacks some reasoning and explanations.

#1. The macroscopic view of the model comprises three components (Fig. 1). However, the arrows used to connect the components are single-sided (oriented). The authors discuss feedback mechanisms between components. But why the authors are omitting some arrows. For example, why there is no feedback mechanism between the population and the decision-making? Democratic governments would be impacted by global changes in the public opinion. Even autocratic governments try to affect the public opinion by means of the controlled media. Similar comments hold true for the missing feedback mechanism between the force engagement and the population.

#2. The connections between decision making agents in Fig. 1 are modelled by means of the networked Kuramoto model (Eq. 1). That is completely understandable, and the authors do elaborate explaining the reasoning behind such an approach. However, the commonsense dictates that the military interaction between the opponents of the conflict should be modelled by the antagonistic terms (from the mathematical point of view). It is widely acceptable to model the antagonistic interaction of conflicting parties by using the multiplicative coupling. This is in stark contrast to the diffusive coupling used to describe the "friendly" synchronization between parties. Typical phenomenological models of the multiplicative and the diffusive coupling can be described by the following simplified differential equations: dX/dt = F(X) + cXY; dY/dt = G(Y) + dXY where X and Y are conflicting populations, t is time, c and d are coupling constants. It is rather surprising that the multiplicative coupling is absent in the model equations introduced in this manuscript.

#3. Another reasoning in favor of the multiplicative coupling is the mathematical theory of cancer dynamics. The interaction between the healthy cells, cancer cells, and the immune system cells (killer cells) is commonly represented by the multiplicative coupling. The theory and dynamics of such systems coupled with the multiplicative coupling is well-developed, studied in detail, and has been applied in numerous clinical cases. The validation of all system parameters, initial and boundary conditions is one of the major issues in any mathematical model of cancer evolution. Meaningless parameters result in meaningless solutions. Again, it is surprising to observe that the authors pay so little attention to the validation of the numerical values of their parameters.

#4. The model of the decision-making networks (Fig. 2) does represent a static picture of the decision-making strategy. It is hard to imagine that the decision-making strategy remains unchanged in the dynamic warfare environment. That could be likely for a wining party - but is difficult to comprehend for the party which does experience several consecutive defeats. The authors could discuss this limitation in their model.

#5. Any mathematical model deserves a proper basic investigation strategy. The necessary and sufficient conditions for the existence of solutions, dynamic equilibria and their stability, limit cycles and conditions of their stability – all that are the basic elements of any modelling strategy. Again, it is surprising to observe that the authors do start their analysis from computational simulations.

#6. The model introduced by the authors in this manuscript is a nonlinear model. Again, it is surprising that the authors do not pay any attention to the possible bifurcation analysis, to the existence of chaotic trajectories and fractal boundary contours.

Reviewer #2: In this article, the authors introduce a model of two combat forces that also interact with a non-combatant population that can support either force. The networked model consists of three dynamic components (force, population, and decision-making) that are defined by the well-known Lanchester, Lotka-Volterra, and Kuramoto-Sakaguchi models. One of the main goals of the paper is to show how the delicate balance between combat initiative and population support determines the victories of either force. The results are shown through numerical simulations that agree with a rigorous mathematical formulation.

The manuscript is well written, the results are clear and convincing, and the mathematical formulation, together with the numerical simulations, supports the main claims. Therefore, I recommend the article for publication in PLOS ONE. Nevertheless, I would like to make some suggestions that the authors could consider in order to improve readability:

1. Some figures, especially Fig. 3, are rather small. Since they include a lot of detail, I recommend increasing the size of the figures. In particular, Fig. 3 could be shown using four vertical panels.

2. Within the paragraph beginning on line 74, the authors include some references related to previous works that have used the Lanchester, Lotka-Volterra, and Kuramoto-Sakaguchi models in the context of conflict environments. It seems a bit odd that most of the references lead to previous works of the authors. To the best of my knowledge, there are many articles in the literature exploring these models, so I encourage the authors to revise the references to give the reader a broader vision of this topic.

6. PLOS authors have the option to publish the peer review history of their article (what does this mean?). If published, this will include your full peer review and any attached files.

Reviewer #1: No

Reviewer #2: No

---

## [Author Response · Author response to Decision Letter 0]

21 Dec 2022

Dear PLOS ONE,

Thank you for considering our work and taking the time to provide comments in an effort to better the manuscript. We have edited the document as requested, with new content given as blue text in the updated manuscript. A point by point explanation of the new content, and how it addresses reviewer comments, is provided below.

Reviewer 1

#1. The macroscopic view of the model comprises three components (Fig. 1). However, the arrows used to connect the components are single-sided (oriented). The authors discuss feedback mechanisms between components. But why the authors are omitting some arrows. For example, why there is no feedback mechanism between the population and the decision-making? Democratic governments would be impacted by global changes in the public opinion. Even autocratic governments try to affect the public opinion by means of the controlled media. Similar comments hold true for the missing feedback mechanism between the force engagement and the population.

A paragraph has been added (lines 85-93) addressing why these choices surrounding model structure and feedback were made, and the limitations they impose. Thank you for bringing this up as transparency around such choices is paramount in such models, but not often addressed.

#2. The connections between decision making agents in Fig. 1 are modelled by means of the networked Kuramoto model (Eq. 1). That is completely understandable, and the authors do elaborate explaining the reasoning behind such an approach. However, the commonsense dictates that the military interaction between the opponents of the conflict should be modelled by the antagonistic terms (from the mathematical point of view). It is widely acceptable to model the antagonistic interaction of conflicting parties by using the multiplicative coupling. This is in stark contrast to the diffusive coupling used to describe the "friendly" synchronization between parties. Typical phenomenological models of the multiplicative and the diffusive coupling can be described by the following simplified differential equations: dX/dt = F(X) + cXY; dY/dt = G(Y) + dXY where X and Y are conflicting populations, t is time, c and d are coupling constants. It is rather surprising that the multiplicative coupling is absent in the model equations introduced in this manuscript.

The multiplicative interaction has a long history in Lanchester models, commonly referred to as the “area/unaimed fire” model. In new text (lines 66-76) we detail the difference and applicability of the two variants, remark why the “aimed fire” variant we use is more applicable, and mention what a mixture of the two is commonly applied for.

#3. Another reasoning in favor of the multiplicative coupling is the mathematical theory of cancer dynamics. The interaction between the healthy cells, cancer cells, and the immune system cells (killer cells) is commonly represented by the multiplicative coupling. The theory and dynamics of such systems coupled with the multiplicative coupling is well-developed, studied in detail, and has been applied in numerous clinical cases. The validation of all system parameters, initial and boundary conditions is one of the major issues in any mathematical model of cancer evolution. Meaningless parameters result in meaningless solutions. Again, it is surprising to observe that the authors pay so little attention to the validation of the numerical values of their parameters.

Additional reasoning surrounding parameter value choices has been given in lines 318-331. Furthermore, discussion surrounding the challenges of validation has been given in lines 332-345. Specifically, we highlight the differences between high fidelity models based on well-established physical laws, and more abstract models which incorporate strategic elements, such as that detailed in the manuscript. Finally, in lines 385-396 we detail how model outputs of Figure 3 can be interpreted against real-world events.

#4. The model of the decision-making networks (Fig. 2) does represent a static picture of the decision-making strategy. It is hard to imagine that the decision-making strategy remains unchanged in the dynamic warfare environment. That could be likely for a wining party - but is difficult to comprehend for the party which does experience several consecutive defeats. The authors could discuss this limitation in their model.

A new paragraph (lines 272-285) has been added to address such limitations of the current model, and how these limitations impact its applicability. The key point is that we seek understanding of the model with a certain level of complexity before adding additional layers, such as dynamical graphs or a campaign level.

#5. Any mathematical model deserves a proper basic investigation strategy. The necessary and sufficient conditions for the existence of solutions, dynamic equilibria and their stability, limit cycles and conditions of their stability – all that are the basic elements of any modelling strategy. Again, it is surprising to observe that the authors do start their analysis from computational simulations.

Solution existence and uniqueness is detailed in lines 298-302. Equilibrium and stability analysis is now given in the first part of Appendix A (lines 694-731). In the main text, model behaviours have been interpreted using this lens in lines 367-370 and 620-621. 

#6. The model introduced by the authors in this manuscript is a nonlinear model. Again, it is surprising that the authors do not pay any attention to the possible bifurcation analysis, to the existence of chaotic trajectories and fractal boundary contours.

Chaos and bifurcation analysis is given in the last part of Appendix A (lines 732-747) – including Figure 10 which is new.

Reviewer 2

1. Some figures, especially Fig. 3, are rather small. Since they include a lot of detail, I recommend increasing the size of the figures. In particular, Fig. 3 could be shown using four vertical panels.

Figures 3, 4 and 5 have been increased in size to address this issue.

2. Within the paragraph beginning on line 74, the authors include some references related to previous works that have used the Lanchester, Lotka-Volterra, and Kuramoto-Sakaguchi models in the context of conflict environments. It seems a bit odd that most of the references lead to previous works of the authors. To the best of my knowledge, there are many articles in the literature exploring these models, so I encourage the authors to revise the references to give the reader a broader vision of this topic.

Lines 8-10 and 102-106 have been added, in addition to the new content brought about through addressing the first reviewer’s comments. In all, 23 references have been added which broaden the reader’s vision on the specific models in question, as well as broader topics such as model validation, modelling philosophies and the impact of abstraction. 

We thank the referees for their constructive feedback and look forward to the decision on our revised paper. 

Kind Regards,

The Authors

---

## [Decision Letter · Decision Letter 1]

17 Jan 2023

`Friend or foe' and decision making initiative in complex conflict environments

PONE-D-22-26713R1

Dear Dr. Zuparic,

We’re pleased to inform you that your manuscript has been judged scientifically suitable for publication and will be formally accepted for publication once it meets all outstanding technical requirements.

Kind regards,

Miguel A. F. Sanjuán

Academic Editor

PLOS ONE

Additional Editor Comments (optional):

Reviewers' comments:

Reviewer's Responses to Questions

**Comments to the Author**

1. If the authors have adequately addressed your comments raised in a previous round of review and you feel that this manuscript is now acceptable for publication, you may indicate that here to bypass the “Comments to the Author” section, enter your conflict of interest statement in the “Confidential to Editor” section, and submit your "Accept" recommendation.

Reviewer #1: All comments have been addressed

Reviewer #2: All comments have been addressed

2. Is the manuscript technically sound, and do the data support the conclusions?

Reviewer #1: Yes

Reviewer #2: Yes

3. Has the statistical analysis been performed appropriately and rigorously? 

Reviewer #1: Yes

Reviewer #2: N/A

4. Have the authors made all data underlying the findings in their manuscript fully available?

Reviewer #1: Yes

Reviewer #2: Yes

5. Is the manuscript presented in an intelligible fashion and written in standard English?

Reviewer #1: Yes

Reviewer #2: Yes

6. Review Comments to the Author

Reviewer #1: The authors did perform a proper revision. All comments and recommendations have been taken into account.

Reviewer #2: (No Response)

7. PLOS authors have the option to publish the peer review history of their article (what does this mean?). If published, this will include your full peer review and any attached files.

Reviewer #1: No

Reviewer #2: No

---

## [Editor Report · Acceptance letter]

24 Jan 2023

PONE-D-22-26713R1 

`Friend or foe' and decision making initiative in complex conflict environments 

Dear Dr. Zuparic:

I'm pleased to inform you that your manuscript has been deemed suitable for publication in PLOS ONE. Congratulations! Your manuscript is now with our production department. 

Kind regards, 

on behalf of

Prof. Miguel A. F. Sanjuán 

Academic Editor

PLOS ONE